# Applications of Modular Co-Design for De Novo 3D Molecule Generation

## Abstract

De novo 3D molecule generation is a pivotal task in drug discovery. However, many recent geometric generative models struggle to produce high-quality 3D structures, even if they maintain 2D validity and topological stability. To tackle this issue and enhance the learning of effective molecular generation dynamics, we present Megalodon–a family of simple and scalable transformer models. These models are enhanced with basic equivariant layers and trained using a joint continuous and discrete denoising co-design objective. We assess Megalodon's performance on established molecule generation benchmarks and introduce new 3D structure benchmarks that evaluate a model's capability to generate realistic molecular structures, particularly focusing on energetics. We show that Megalodon achieves state-of-the-art results in 3D molecule generation, conditional structure generation, and structure energy benchmarks using diffusion and flow matching. Furthermore, we demonstrate that scaling Megalodon produces up to 49x more valid molecules at large sizes and 2-10x lower energy compared to the prior best generative models.

## 1 Introduction

Molecular Generative models have been heavily explored due to the allure of enabling efficient virtual screening and targeted drug design (Gómez-Bombarelli et al., 2018). Similar to the rise in their application to computer vision (CV) (Peebles & Xie, 2022; Esser et al., 2024; Ma et al., 2024), Diffusion and Flow Matching models have been applied for tasks including molecule design, molecular docking, and protein folding (Schneuing et al., 2022; Corso et al., 2023a; Abramson et al., 2024). Across CV and chemical design, the scaling of model architectures and training data have seen significant accuracy improvements but questions surrounding how to scale effectively still persist (Corso et al., 2024; Durairaj et al., 2024).

Specifically for 3D molecule generation (3DMG), where the task is to unconditionally generate valid and diverse 3D molecules, diffusion models have shown great promise in enabling accurate generation starting from pure noise (Hoogeboom et al., 2022). The iterative nature of diffusion models allows them to explore a diverse range of molecular configurations, ideally providing valuable insights into potential drug candidates and facilitating the discovery of novel compounds. However, unlike in CV, which has seen systematic evaluations of training data and scaling, with tangible benchmark results (Esser et al., 2024), measuring success in de novo molecule generation is quite difficult. As a result, there is a nonlinear path to determining what truly is making an impact if, in each model, the data, architecture, training objective, and benchmarks differ. Furthermore, the commonly shared 3DMG benchmarks that do exist only evaluate 2D quantities, ignoring 3D structure, conformational energy, and model generalization to large molecule sizes–all quantities that are imperative for real-world use. In this work, we explore the above in the context of 3DMG and its interpretable benchmarks to directly target larger molecules and model scaling.

Our main contributions are as follows:

- We present Megalodon, a simple and scalable transformer-based architecture for multi-modal molecule diffusion and flow matching. This is the first 3DMG model to be tested with both objectives, with both obtaining state-of-the-art results. We show that our diffusion model excels at structure and energy benchmarks, whereas our flow matching model yields better 2D stability and the ability to use 25x fewer inference steps than its diffusion counterpart.

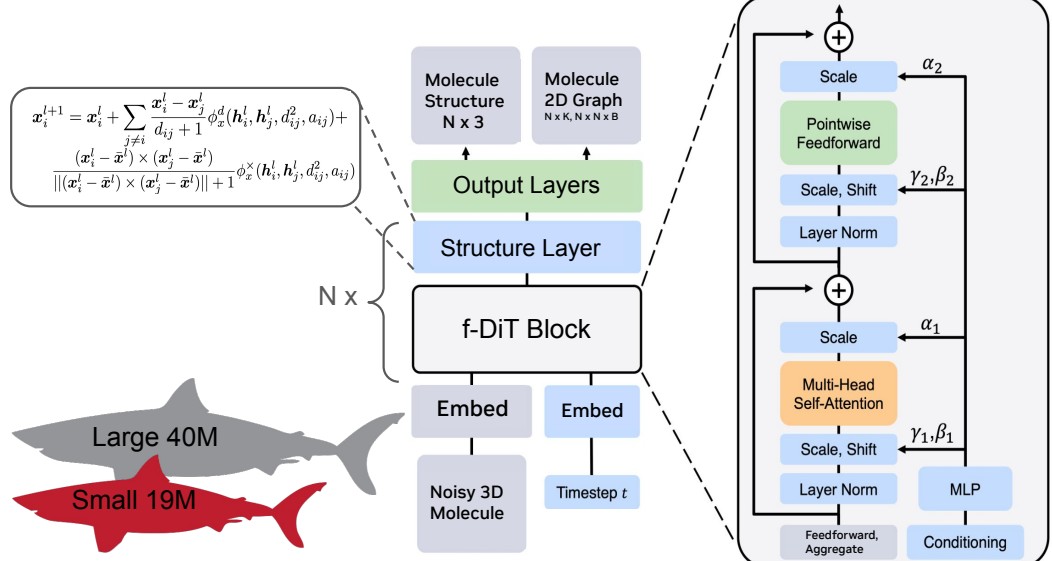

Figure 1: Megalodon Architecture: molecules are separated into 3D structures and discrete atom types, bond types, and atom charge features. All features are embedded separately, passed through a feed-forward layer, and aggregated to produce the input tokens for the fused DiT blocks. The embedded structure features and DiT outputs are passed to an EGNN-based layer to refine the structure prediction. The output heads consist of standard MLPs and an EGNN layer for bond refinement. Note a skip connection exists between structure layers as shown in the update equation.

- Megalodon is the first model capable of unconditional molecule generation and conditional structure generation without retraining or finetuning.
- We introduce new geometric benchmarks focusing on the interpretable physics-based quantum mechanical (QM) and molecular conformational energy.

## 2 BACKGROUND

### 2.1 3D MOLECULE GENERATION

In de novo 3D molecule generation (3DMG), a molecule's 3D structure and 2D topology are simultaneously generated. We define a molecule $M = (X, H, E, C)$ with $N$ atoms where $X \in \mathbf{R}^{N \times 3}$, $H \in \{0, 1\}^{N \times A}$, $E \in \{0, 1\}^{N \times N \times B}$, and $C \in \{0, 1\}^{N \times K}$ represents the atom coordinates, element types, bond types adjacency matrix, and formal charges respectively. $X$ is modeled as a continuous variable whereas $H$, $E$, and $C$ are discrete one-hot variables.

### 2.2 IMPORTANT QUALITIES OF 3D MOLECULES

The GEOM dataset (Axelrod & Gómez-Bombarelli, 2022) is widely used for 3D molecular structure (conformer) generation tasks, containing 3D conformations from both the QM9 and drug-like molecule (DRUGS) databases, with the latter presenting more complex and realistic molecular challenges. Conformers in the dataset were generated using CREST (Pracht et al., 2024), which performs extensive conformational sampling based on the semi-empirical extended tight-binding method (GFN2-xTB) (Bannwarth et al., 2019). This ensures that each conformation represents a local minimum in the GFN2-xTB energy landscape.

Energy, in the context of molecular conformations, refers to the potential energy of a molecule's structure, which is a key determinant of its stability. Lower-energy conformations are typically more stable and are found at the minima on the potential energy surface (PES). For a generative model to succeed, it must not only generate molecules that are chemically valid but also ones that correspond to low-energy conformations, reflecting local minima on the PES. Thus, energy serves as the ultimate measure of success in molecular modeling, as it directly correlates with the physical realism and stability of the generated structures.

A key requirement for generative models is their ability to implicitly learn this energy landscape and produce molecules that are local minima of the potential energy surface. However, since GFN2-xTB is itself a model rather than a universal energy function, comparing energies across different potentials (e.g., using GFN2-xTB optimized structures but computing energies with MMFF (Halgren, 1996)) can introduce systematic errors. Differences in potential models, such as optimal bond lengths, may lead to unreliable results. Overall, the goal of 3DMG is to generate valid and low-energy molecules mimicking the energy landscape of the GEOM dataset.

## 2.3 RELATED WORK

Hoogeboom et al. (2022) first introduced continuous diffusion modeling for coordinates and atom types using a standard EGNN architecture (Satorras et al., 2021). Following this, many models have been produced that make slight changes to the architecture and interpolant schedule to generate atom coordinates and types (Song et al., 2023). While initially effective, they rely on the molecule-building software OpenBabel (O'Boyle et al., 2011) to infer and update the bond locations and types, which is a critical aspect of the stability calculations. The issues and biases of OpenBabel have been heavily explored, and as a result, methods began to generate the bond locations and types in the generative process (Walters, 2024). Vignac et al. (2023) was the first to use continuous diffusion for coordinates and discrete diffusion for the atom and bond types, removing the OpenBabel requirement. Le et al. (2024) used the same training objective but introduced a more effective equivariant architecture. Recently Irwin et al. (2024) uses continuous and discrete flow matching with a latent equivariant graph message passing architecture to show improved performance. For further discussion please see §B

## 2.4 STOCHASTIC INTERPOLANTS

**Continuous Gaussian Interpolation** Following (Lipman et al., 2023; Albergo et al., 2023), in the generative modeling setting, we construct interpolated states between an empirical data and a Gaussian noise distribution $\mathcal{N}(\mathbf{x}_t; \beta(t)x_1, \alpha(t)^2 \boldsymbol{I})$, this is,

$$\mathbf{x}_t = \alpha(t)\boldsymbol{\epsilon} + \beta(t)\mathbf{x}_1, \tag{1a}$$

$$\mathbf{x}_1 = \frac{\mathbf{x}_t - \alpha(t)\boldsymbol{\epsilon}}{\beta(t)} \tag{1b}$$

where $\boldsymbol{\epsilon} \sim \mathcal{N}(\boldsymbol{\epsilon}; \mathbf{0}, \boldsymbol{I})$ and $\mathbf{x}_1 \sim p_{\text{data}}(\mathbf{x}_1)$. Common choices for the interpolation include (assuming $t \in [0, 1]$), with $t = 1$ corresponding to data and $t = 0$ to noise:

- Variance-preserving SDE-like from the diffusion model literature (Song et al., 2021): $\alpha(t) = \sqrt{1 - \gamma_t^2}$ and $\beta(t) = \sqrt{\gamma_t^2}$ with some specific "noise schedule" $\gamma_t$ which is commonly written as $\sqrt{\overline{\alpha_t}}$ from Ho et al. (2020).

- Conditional linear vector field (Lipman et al., 2023): $\alpha(t) = 1 - (1 - \sigma_{\min})t$ and $\beta(t) = t$ with some smoothening of the data distribution $\sigma_{\min}$.

**Continuous Diffusion** Continuous Denoising Diffusion Probabilistic Models (DDPM) integrate a gradient-free forward noising process based on a predefined discrete-time variance schedule ( Eq. 1a) and a gradient-based reverse or denoising process (Ho et al., 2020). The denoising model can be parameterized by data or noise prediction as they can be equilibrated via Eq. 1b. Following Le et al. (2024), we use the following training objective and update rule:

$$\mathcal{L}_{\text{DDPM}}(\theta) = \mathbb{E}_{t, \boldsymbol{\epsilon} \sim \mathcal{N}(\boldsymbol{\epsilon}; \mathbf{0}, \boldsymbol{I}), \mathbf{x}_1 \sim p_{\text{data}}(\mathbf{x}_1)} ||\mathbf{x}_\theta(t, \mathbf{x}_t) - \mathbf{x}_1||^2 \tag{2}$$

$$\mu_\theta(t, \mathbf{x}_t) = \mathbf{f}(\alpha(t), \beta(t)) * \mathbf{x}_\theta(t, \mathbf{x}_t) + \mathbf{g}(\alpha(t), \beta(t)) * \mathbf{x}_t$$
$$\mathbf{x}_{t+1} = \mu_\theta(t, \mathbf{x}_t) + \sigma((\alpha(t), \beta(t)) * \boldsymbol{\epsilon} \tag{3}$$

where functions $\mathbf{f}$, $\mathbf{g}$, and $\boldsymbol{\sigma}$ are defined for any noise schedule such as the cosine noise schedule used in Vignac et al. (2023).

**Continuous Flow Matching**    Flow matching (FM) models are trained using the conditional flow matching (CFM) objective to learn a time-dependent vector field $\mathbf{v}_\theta(t, \mathbf{x}_t)$ derived from a simple ordinary differential equation (ODE) that pushes samples from an easy-to-obtain noise distribution to a complex data distribution.

$$\mathcal{L}_{\text{CFM}}(\theta) = \mathbb{E}_{t,\boldsymbol{\epsilon}\sim\mathcal{N}(\boldsymbol{\epsilon};\mathbf{0},\boldsymbol{I}),\mathbf{x}_1\sim p_{\text{data}}(\mathbf{x}_1)} ||\mathbf{v}_\theta(t, \mathbf{x}_t) - \frac{d}{dt}\mathbf{x}_t||^2$$

$$= \mathbb{E}_{t,\boldsymbol{\epsilon}\sim\mathcal{N}(\boldsymbol{\epsilon};\mathbf{0},\boldsymbol{I}),\mathbf{x}_1\sim p_{\text{data}}(\mathbf{x}_1)} ||\mathbf{v}_\theta(t, \mathbf{x}_t) - \dot{\alpha}(t)\boldsymbol{\epsilon} - \dot{\beta}(t)\mathbf{x}_1||^2, \tag{4}$$

The time-differentiable interpolation seen in Eq. 1a gives rise to a probability path that can be easily sampled. For more details on how to relate the Gaussian diffusion and CFM objectives with the underlying score function of the data distribution, please see Appendix A.

In practice, many methods use a "data prediction" objective to simplify training, which gives rise to the following loss function and inference Euler ODE update step following the conditional linear vector field (Lipman et al., 2023; Irwin et al., 2024).

$$\mathcal{L}_{\text{CFM}}(\theta) = \mathbb{E}_{t,\boldsymbol{\epsilon}\sim\mathcal{N}(\boldsymbol{\epsilon};\mathbf{0},\boldsymbol{I}),\mathbf{x}_1\sim p_{\text{data}}(\mathbf{x}_1)} ||\mathbf{x}_\theta(t, \mathbf{x}_t) - \mathbf{x}_1||^2 \tag{5}$$

$$\mathbf{v}_\theta(t, \mathbf{x}_t) = \frac{\mathbf{x}_\theta(t, \mathbf{x}_t) - x_t}{1 - t},$$

$$\mathbf{x}_{t+1} = x_t + \mathbf{v}_\theta(t, \mathbf{x}_t)dt \tag{6}$$

**Discrete Diffusion**    Following Austin et al. (2021), Discrete Denoising Diffusion Probabilistic Models (D3PMs) apply the same concept as continuous diffusion but over a discrete state space. Like the continuous counterpart that relies on a predefined schedule to move mass from the data to prior distribution, D3PM uses a predefined transition matrix that controls how the model transitions from one discrete state to another.

For scalar discrete random variables with $K$ categories $x_t, x_{t-1} \in 1, ..., K$ the forward transition probabilities can be represented by matrices: $[Q_t]_{ij} = q(x_t = j | x_{t+1} = i)$. Starting from our data $x_1$ or $x_T$ (where $T$ is the total number of discrete time steps[1], we obtain the following $T - t + 1$ step marginal and posterior at time $t$:

$$q(x_t|x_{t+1}) = \text{Cat}(x_t; p = x_{t+1}Q_t), \quad q(x_t|x_T) = \text{Cat}\left(x_t; p = x_T\overline{Q}_t\right), \quad \text{with} \quad \overline{Q}_t = Q_t Q_{t+1} \ldots Q_T$$

$$q(x_{t+1}|x_t, x_T) = \frac{q(x_t|x_{t+1}, x_T)q(x_{t+1}|x_T)}{q(x_t|x_T)} = \text{Cat}\left(x_{t+1}; p = \frac{x_t Q_t^\top \odot x_T\overline{Q}_{t+1}}{x_T\overline{Q}_t x_t^\top}\right) \tag{7}$$

Here $Q$ is defined as a function of the same cosine noise schedule used in continuous DDPM such that the discrete distribution converges to the desired terminal distribution (*i.e.* uniform prior) in T discrete steps. Similar to the use of mean squared error loss for DDPM, D3PM uses a discrete cross-entropy objective between the true and predicted discrete data.

**Discrete Flow Matching**    Following Campbell et al. (2024), we use the Discrete Flow Matching (DFM) framework to learn conditional flows for the discrete components of molecule generation ( atom types, bond types, and atom charges). We use the following DFM interpolation in continuous time, where $S$ is the size of the discrete state space:

$$p_{t|1}^{\text{unif}}(x_t|x_1) = q(x_t|x_1) = \text{Cat}(t\delta\{x_1, x_t\} + (1 - t)\frac{1}{S}), \tag{8}$$

Similar to discrete diffusion, we use the cross-entropy objective for training. Please see Campbell et al. (2024) for sampling procedure details.

**Diffusion vs. Flow Matching**    We see that for both Diffusion and CFM, the loss functions used in practice are identical. Differences arise in how we build the interpolation, how we sample from these models, and their theoretical constraints. Diffusion models rely on complex interpolation schedules that are tuned to heavily weight the data distribution using a uniform time distribution. In

---

[1]We adjust the direction of time for diffusion to match the FM equations such that T=1 is data.

contrast, FM commonly uses a simple linear interpolation but can achieve that same data distribution weighting by sampling from more complex time distributions. The choices of time distributions and interpolation schedules can be chosen appropriately to make FM and Diffusion equivalent in the Gaussian setting (see Sec. A). We show in Fig. 2 the interpolation and time distribution differences that mimic the same weighting of $p_{\text{data}}$ at T=1 that are currently used in recent 3DMG models (Le et al., 2024; Irwin et al., 2024).

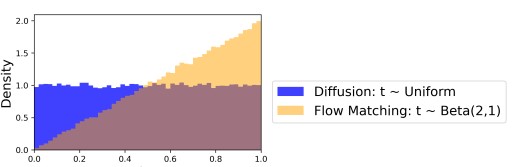
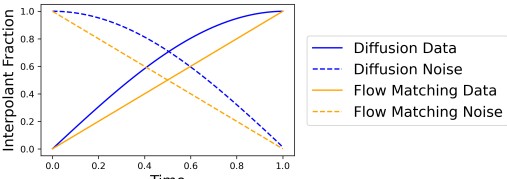

(a) Time distributions used for molecule generation     (b) FM linear vs. Diffusion cosine interpolant

Figure 2: Time and interpolation comparison between Megalodon and Megalodon-flow

Diffusion models inherently rely on simulating Gaussian stochastic processes. In the forward process, data points are progressively noised, converging towards a Gaussian distribution. This process, derived from score-based generative models, aims to learn the score function (the gradient of the data distribution's log density) to reverse the diffusion process. The generative model effectively solves a Stochastic Differential Equation (SDE) that describes how data diffuses towards noise and how it can be denoised in reverse. The reverse process requires SDE simulation at every step, which involves sampling from a learned probabilistic model that estimates how to remove noise. This involves simulating random variables at each time step, making diffusion models highly dependent on repeated stochastic simulation.

Flow Matching, on the other hand, learns a continuous vector field that deterministically "flows" one distribution to another. The model learns this flow by matching the velocity field that pushes samples from a source distribution to a target distribution. Once the vector field is learned, generating samples involves solving an ODE that defines a continuous and deterministic trajectory from the source to the target distribution. Unlike diffusion models, which require simulating a series of stochastic transitions (noising and denoising) over many steps, flow matching learns a single, continuous flow. Sampling involves solving an ODE (or, in some cases, a deterministic SDE with noise) to move from the base distribution to the target in a smooth, deterministic fashion.

For DDPM, the equations only hold for the Gaussian path with access to a well-formed score function. This is why techniques like mini-batch Optimal Transport (OT) can be applied to FM but not Diffusion to align $p_{\text{data}}$ and $p_{\text{ref}}$ (Tong et al., 2023). In FM, the vector field is learned, which, in the absence of OT, can be derived as a function of the score function, but having access to the score function is not a requirement to sample in a simulation-free or deterministic way.

## 3  METHODS

**Megalodon Architecture**  Since 3DMG allows for the simultaneous generation of a discrete 2D molecular graph and its 3D structure, we intentionally designed our architecture with a core transformer trunk to better model discrete data (Vaswani, 2017; Brown et al., 2020). Fig. 1 illustrates the model architecture, which is comprised of N augmented transformer blocks followed by linear layers for discrete data projection.

First, the input structure, atom types, and bond types are fused and aggregated to create a single molecule feature. This is passed into the standard multi-head attention module with adaptive layernorm. The scaled output is then passed into separate adaptive layernorm feedforward blocks for the atom types and bond types. We augment the DiT block defined in Peebles & Xie (2022) to take in structure, atom types, and bond types to produce updated atom and bond types, which are then passed into a simple structure layer. The structure layer only updates the predicted structure via a standard distance-based EGNN update with a cross-product term (Satorras et al., 2021; Schneuing et al., 2022). We emphasize that this cross-product term is critical for model performance. At a high level, the transformer block updates our discrete data, and our equivariant layer updates our structure. Megalodon uses standardized scaling tricks such as query key prenorm and equivariant norm in

our structure layer (Hayes et al., 2024). We also note Megalodon, at 4x more parameters, is more memory efficient than EQGAT-diff (Le et al., 2024), enabling 2x larger batch size while still having the quadratic dependency of fully connected edge features. For more details, please see Sec. C.

We introduce a generative scaling benchmark, and as we show, the performance of 3DMG models is correlated with the size of the generated molecules. We note that our large model is, in fact, not that large compared to recent biological models Lin et al. (2023); Hayes et al. (2024) and can be further scaled beyond 40M params if further benchmarks are developed.

**Training Objective**  We explore Megalodon in the context of diffusion and flow matching. For our diffusion flavored model, following Le et al. (2024); Vignac et al. (2023) we use the same weighted cosine noise schedules, DDPM, and discrete D3PM objective. When using conditional flow matching, we apply the same training objective and hyperparameters as Irwin et al. (2024), including equivariant optimal transport. In this way, for diffusion and flow matching, we train and evaluate our model in an *identical way* including hyperparameters to prior models of the respective types.

In our experiments with EQGAT-diff, we found that the diffusion objective with data-like priors possesses an interesting but potentially harmful behavior. Although the noise sample from the data-prior and the true data sample have bonds, the model consistently generates no bonds for all time $\leq 0.5$, which corresponds to an interpolation with $\leq 70\%$ of the data as seen in Fig. 2(b). Therefore there is no useful information for the edge features in half the training and inference samples. As a result, only when the structure error is low, as the model starts with 70% data in the interpolation, does the bond prediction accuracy jump to near-perfect accuracy. Thus, only when the structure is accurate was the 2D graph accurate, which is counterintuitive to the independent and simultaneous objective. In other words, the 2D graph does not inform the 3D structure as one would expect to happen, and we would want equal importance on the 2D topology and 3D structure.

To address this inefficiency, as the structure, atom type, and bond type prediction inform each other to improve molecule generation, we introduce a subtle change to the training procedure similar to Campbell et al. (2024). Keeping each data type having its own independent noise schedule, we enable a concrete connection between the discrete and continuous data that it is modeling. Explicitly, rather than sampling a single time variable, we introduce a second noise variable to create $t_{continuous}$ and $t_{discrete}$, both sampled from the same time distribution. Now discrete and continuous data are interpolated with their respective time variable *and* maintain the independent weighted noise schedules. We note that the MiDi weighted cosine schedules were already adding different levels of noise for the same time value. Now, we take that one step further and allow the model to fill in the structure given the 2D graph and learn to handle more diverse data interpolations.

**Self Conditioning**  Following Chen et al. (2022), we train Megalodon with self-conditioning similar to prior biological generative models (Yim et al., 2023; Stärk et al., 2024; Irwin et al., 2024). We found that constructing self-conditioning as an outer model wrapper with a residual connection led to faster training convergence:

$$
\begin{aligned}
x_{\text{sc}} &= \text{model}(x_t) \\
x_t &= \text{MLP}([x_{\text{sc}}, x_t]) + x_t \\
x_{\text{pred}} &= \text{model}(x_t)
\end{aligned}
\tag{9}
$$

Specifically for 3DMG, self-conditioning is applied independently to each molecule component $M = (X, H, E, C)$, where the structure component uses linear layers without bias and all discrete components operate over the raw logits rather than the one-hot predictions.

## 4  EXPERIMENTS

**Data**  GEOM Drugs is a dataset of drug-like molecules with an average size of around 44 atoms (Axelrod & Gómez-Bombarelli, 2022). Following Le et al. (2024), we use the same training splits as Vignac et al. (2023). We emphasize that the traditional metrics are calculated by first sampling molecule sizes from the dataset( Fig. 5) and then generating molecules with the sampled number of atoms, including explicit hydrogens. We show in Sec. 4.1 that this does not illustrate the full generative capacity, as in many real-world instances, people want to generate molecules with greater than 100 atoms (Békés et al., 2022).

Table 1: Measuring Unconditional Molecule Generation: 2D topological and 3D distributional benchmarks. * Denotes taken from EQGAT-Diff.

| Model | Steps | 2D Topological (↑) | | | 3D Distributional (↓) | |
|---|---|---|---|---|---|---|
| | | Atom Stab. | Mol Stab. | Validity | Bond Angle | Dihedral |
| EDM+OpenBabel* | 1000 | 0.978 | 0.403 | 0.363 | – | – |
| MiDi* | 500 | 0.997 | 0.897 | 0.705 | – | – |
| EQGAT-diff$_{disc}^{x0}$ | 500 | 0.998 | 0.935 | 0.830 | 0.858 | 2.860 |
| EGNN + cross product | 500 | 0.982 | 0.713 | 0.223 | 14.778 | 17.003 |
| Megalodon-small | 500 | 0.998 | 0.961 | 0.900 | 0.689 | 2.383 |
| Megalodon | 500 | **0.999** | **0.977** | **0.927** | **0.461** | **1.231** |
| SemlaFlow | 100 | **0.998** | 0.979 | 0.920 | 1.274 | **1.934** |
| Megalodon-flow | 100 | 0.997 | **0.990** | **0.948** | **0.976** | 2.085 |

## 4.1 Unconditional De Novo Generation

**Problem Setup**   Following Le et al. (2024) we generate 5000 molecules (randomly sampling the number of atoms from the train distribution see Fig. 5), and report (1) Atom Stability: the percentage of individual atoms that have the correct valency according to its electronic configuration that was predefined in a lookup table, (2) Molecule Stability: percentage of molecules in which all atoms are stable, (3) Connected Validity: fraction of molecules with a single connected component which can be sanitized with RDKit. We also introduce two structural distributional metrics for the generated data: (4) bond angles and (5) dihedral angles, calculated as the weighted sum of the Wasserstein distance between the true and generated angle distributions, with weights based on the central atom type for bond angles and the central bond type for dihedral angles, respectively. We note this is a high-level metric and not reducible to a physically meaningful per-molecule error. To combat this, please see our new physical structure metrics in Sec. 4.3.

**Baselines**   EQGAT-diff has 12.3M parameters and leverages continuous and discrete diffusion (Le et al., 2024). SemlaFlow has 23.3M params[2] and is trained with conditional flow matching with equivariant optimal transport (Irwin et al., 2024). We report two Megalodon sizes, small (19M) and large (40.6M). We train with identical objectives and settings to both EQGAT-diff and SemlaFlow. We also compare to older diffusion models, including MiDi and EDM, as they introduce imperative techniques from which the more recent models are built.

**Analysis**   Both the diffusion and flow matching versions of Megalodon achieve state-of-the-art results. With the FM version obtaining better topological accuracy and the diffusion version seeing significantly improved structure accuracy. This experiment shows that the underlying augmented transformer is useful for the discrete and continuous data requirements of 3DMG, regardless of the interpolant and sampling methodology. We also see that the transformer part is crucial for Megalodon's success as just using the EGNN with cross-product updates with standard edge and feature updates for the non-equivariant quantities performs quite poorly. We also note that all methods obtain 100% *uniqueness*, 88-90% *diversity*, and 99% *novelty* following (Le et al., 2024) definitions with no meaningful performance differences. For further model comparisons and model ablations surrounding reducing the number of inference steps please see Appendix. Table 5.

**Impact of molecule size on performance**   As Table 1 shows average results over 5000 molecules of relatively small and similar sizes, it is hard to understand if the models are learning how to generate molecules or just regurgitating training-like data. We design an experiment to directly evaluate this question and see how models perform as they are tasked to generate molecules outside the support region of the train set. We see in Fig. 3 that the topological model performance is a function of length (for full-size distribution, see Fig. 5). Here for each length [30, 125] we generate 100 molecules and report the percentage of stable and connected valid molecules.

We emphasize that Table 1 illustrates only a slice of the performance via the average of 5K molecules sampled from the train set size distribution. We note that although molecules with greater than 72

---

[2]Checkpoint from public code has 2 sets of 23.2M params, one for the last gradient step and EMA weights

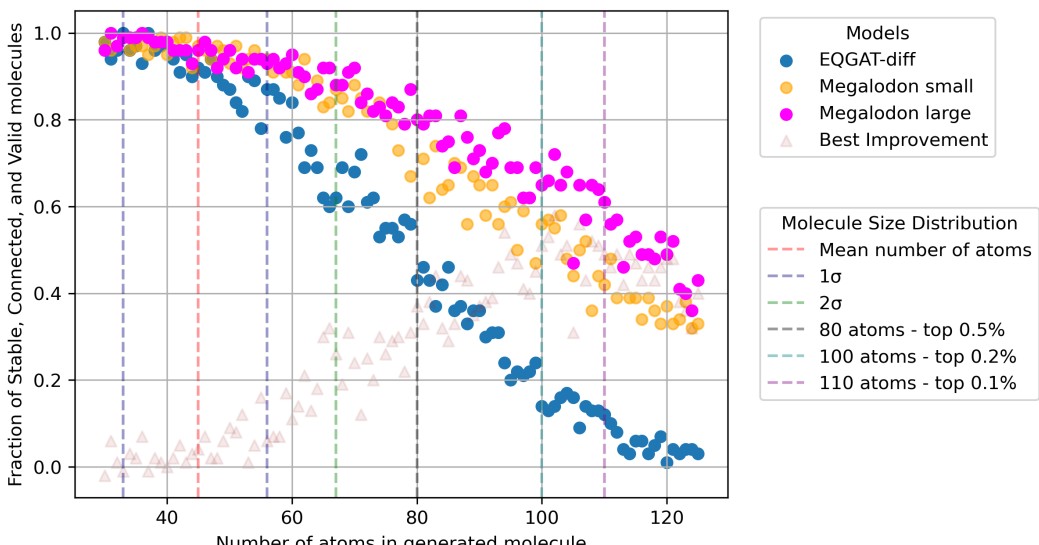

Figure 3: Diffusion model performance as a function of molecule size.

atoms make up $\leq 1\%$ of the train set, Megalodon demonstrates roughly 2-49x better performance than EQGAT-diff for the larger half of the generated molecule sizes. We hypothesize that since molecule stability is a discrete 2D measurement, the transformer blocks in Megalodon allow it to better generalize even if seeing similar molecules in less than 0.1% of the training data. In other words, the ability of transformers to excel at modeling discrete sequential data improves our generative performance. We want to point out that all tested models are trained with identical datasets, hyperparameters, diffusion schedules, and training objectives. The only difference is the architecture. We also see that the ability to scale our simple architecture allows the model to even better generate molecules outside the region of data support. Lastly, we chose to focus on only the diffusion models here as they exhibit the best structure benchmark performance.

## 4.2 CONDITIONAL STRUCTURE GENERATION

Similar to the 3D molecule generation task, we use the GEOM-Drugs dataset to evaluate the conditional structure generation capabilities of our model. Given all unconditional 3DMG models are trained with independent noising of coordinates, atoms, and, in some cases, bonds, we want to evaluate how accurate the structural component is. We note this is something that is lacking from the existing prior benchmarks, as when generating novel de novo molecules, there is no ground truth structure to compare against. In the task of conditional structure generation, all models are given the molecule 2D graph (atom types, bonds) and asked to generate the 3D structure in which ground truth data exists. Given Vignac et al. (2023) and Jing et al. (2022) use different train/test splits, we evaluate all methods on the overlap of 200 held-out molecules, with all methods generating 43634 structures in total. Due to the similarities with the baselines and its superior unconditional structure accuracy, we compare Megalodon trained with diffusion against well-performing methods with public reproducible code.

**Problem setup.** We report the average minimum RMSD (AMR) between ground truth and generated structures, and Coverage for Recall and Precision. Coverage is defined as the percentage of conformers with a minimum error under a specified AMR threshold. Recall matches each ground truth structure to its closest generated structure, and Precision measures the overall spatial accuracy of each generated structure. Following Jing et al. (2022), we generate two times the number of ground truth structures for each molecule. More formally, for $K = 2L$, let $\{C_l^*\}_{l \in [1,L]}$ and $\{C_k\}_{k \in [1,K]}$ respectively be the sets of ground truth and generated structures:

$$\text{COV-Precision} := \frac{1}{K} \left| \left\{ k \in [1..K] : \min_{l \in [1..L]} \text{RMSD}(C_k, C_l^*) < \delta \right\} \right|,$$

$$\text{AMR-Precision} := \frac{1}{K} \sum_{k \in [1..K]} \min_{l \in [1..L]} \text{RMSD}(C_k, C_l^*),$$

(10)

Table 2: Quality of ML generated conformer ensembles for GEOM-DRUGS ($\delta = 0.75$Å) test set in terms of Coverage (%) and Average RMSD (Å). Bolded results are the best, and the underlined results are second best.

| | Recall | | | | Precision | | | |
| | Coverage ↑ | | AMR ↓ | | Coverage ↑ | | AMR ↓ | |
| Method | Mean | Med | Mean | Med | Mean | Med | Mean | Med |
|---|---|---|---|---|---|---|---|---|
| GeoDiff | 42.1 | 37.8 | 0.835 | 0.809 | 24.9 | 14.5 | 1.136 | 1.090 |
| Torsional Diffusion | **75.3** | **82.3** | **0.569** | **0.532** | 56.5 | 57.9 | 0.778 | 0.731 |
| EQGAT-Diff | 0.8 | 0.0 | 2.790 | 2.847 | 0.1 | 0.0 | 3.754 | 3.771 |
| Megalodon | 71.4 | 75.0 | 0.573 | 0.557 | **61.2** | **63.1** | **0.719** | **0.696** |

where $\delta$ is the coverage threshold. The recall metrics are obtained by swapping ground truth and generated conformers.

**Baselines** We compare Megalodon with EQGAT-Diff, GeoDiff (Xu et al., 2022), and TorsionalDiffusion (Jing et al., 2022). For the unconditional 3DMG models, including Megalodon, we prompt them with the ground truth atom types and bond types to guide the generation of the structure along the diffusion process. This is done by replacing the input and output with the fixed conditional data. We do this to assess what the model is actually learning across the multiple data domains. The central question being, is the model learning how to generate molecules over the spatial and discrete manifolds, or is it just learning how to copy snapshots of training-like data?

**Analysis** We see in Table 2 that EQGAT-diff is unable to generate any remotely valid structures. Even though all modalities are being denoised independently at different rates, the model cannot generate the structure given ground truth 2D molecule graphs. This is also seen during the sampling process, where diffusion models trained with similar denoising objectives as EQGAT-diff generate no bonds until the structure has seemingly converged. Therefore during most of the sampling process, the edge features which make up a large portion of the computational cost hold no value.

In comparison, Megalodon generates structures with competitive precision and recall by building a relationship between the discrete and continuous data directly in the training process described in §3. Half the time all data types are independently noised as normal with their respective time variables and schedules, the other half we only add noise to the structure. Therefore, our model learns to build a relationship between true 2D graphs and their 3D structure, as well as any interpolation between the three data tracks that are interpolated independently with different schedulers.

Megalodon demonstrates that its unconditional discrete diffusion objective is crucial for its conditional performance. In other words, the discrete diffusion training objective improves the conditional continuous generative performance. This is evident in the comparison between GeoDiff and Megalodon. GeoDiff is trained on the same conditional Euclidean structure objective as Megalodon (with similar EGNN-based architecture) with 10x more diffusion steps, with both models taking in identical inputs. We see that since Megalodon is able to generate molecules from pure noise, it better learns structure and as a result can be prompted to generate accurate structures.

Interestingly, compared to Torsional Diffusion, which initializes the 3D structure with an expensive RDKit approximation to establish all bond lengths and angles and then only modifies the dihedral angles, we see quite competitive performance. Before, it was understood that by restricting the degrees of freedom with good RDKit structures, the performance jump from GeoDiff to Torsional Diffusion was observed. Now we see that with the same euclidean diffusion process, similar accuracy improvements can be gained by learning how to generate accurate discrete molecule topology via discrete diffusion. We want to note that there have been recent advances on top of Torsional Diffusion (Corso et al., 2023b) and other conformer-focused models that are not public (Wang et al., 2023). We use this benchmark more to analyze the underlying multi-modal diffusion objective and focus on the underlying model comparisons. Megalodon is not a conformer generation model but a molecule generation model capable of de novo and conditional design. Overall, Megalodon shows how independent time interpolation and discrete diffusion create the ability for the model to be prompted or guided with a desired 2D topology to generate accurate 3D structures.

Table 3: xTB Relaxation Error: Length Å, angles degrees, energy kcal/mol. These metrics are taken over the valid molecules from Table 1. Methods are grouped by model type: diffusion (500 steps) and flow matching (100 steps)

| Model | Bond Length | Bond Angles | Dihedral | Median $\Delta E_{\text{relax}}$ | Mean $\Delta E_{\text{relax}}$ |
|---|---|---|---|---|---|
| GEOM-Drugs | 0.0000 | 0.00 | 7.2e-3 | 0.00 | 1.0e-3 |
| EQGAT-diff | 0.0076 | 0.95 | 7.98 | 6.36 | 11.06 |
| Megalodon-small | 0.0085 | 0.88 | 7.28 | 5.78 | 9.74 |
| Megalodon | **0.0061** | **0.66** | **5.42** | **3.17** | **5.71** |
| SemlaFlow | 0.0309 | 2.03 | 6.01 | 32.96 | 93.13 |
| Megalodon-flow | **0.0225** | **1.59** | **5.49** | **20.86** | **46.86** |

### 4.3 UNCONDITIONAL STRUCTURE-BASED ENERGY BENCHMARKS

**Problem setup**   Each ground truth structure in GEOM dataset represents a low-energy conformer within its ensemble, highlighting two key aspects. First, these molecules are local minima on the GFN2-xTB potential energy surface. Second, their energies are lower compared to other conformations sampled in the ensemble. Previously, these quantities have not been thoroughly evaluated for generated molecules. To address this gap, we directly measure how closely a generated molecule approximates its nearest local minimum (i.e., its relaxed structure). We measure the energy difference between the initial generated structure and its relaxed counterpart, as well as structural changes in bond lengths, bond angles, and dihedral (torsion) angles. This approach allows us to evaluate the ability of generative models to produce molecules that are true local minima, facilitating faster ranking of generated structures without additional minimization steps.

**Analysis**   We see that for both diffusion and flow matching, Megalodon is better than its prior counterparts. Overall, Megalodon trained with diffusion performs best with roughly 2-10x lower median energy when compared to prior generative models. Notably, our model's median relaxation energy difference $\Delta E_{\text{relax}}$ is around 3 kcal/mol, which approaches the thermally relevant interval of 2.5 kcal/mol (Axelrod & Gómez-Bombarelli, 2022). Megalodon is the first method to achieve such proximity to this thermodynamic threshold, marking a significant milestone in 3D molecular generation. For more details and justification, please see Appendix §E.

We note that while the loss function between FM and diffusion is identical in this instance, we see both flow models have an order of magnitude larger bond angle error, which translates to a similar energy performance gap. The xTB energy function is highly sensitive to bond lengths; small deviations in bond lengths can lead to significant increases in energy due to the steepness of the potential energy surface in these dimensions. A precise representation of bond lengths is crucial because inaccuracies directly impact the calculated energy, making bond length errors a primary contributor to higher relaxation energies in flow models. We hypothesize that since the Flow models scale the input structures to have a variance of 1 to make matching the Gaussian prior easier, we lower the local spatial precision necessary for bond length and angle generation (Irwin et al., 2024).

## 5 CONCLUSIONS

Megalodon enables the accurate generation of de novo 3D molecules with both diffusion and flow matching. We show with a simple and scalable augmented transformer architecture that significant improvements are gained, especially when generating outside the region of support for the training distribution as it pertains to molecule sizes. Megalodon also demonstrates the ability to achieve great accuracy in conditional structure generation due to being trained to generate complete molecules from scratch. We also introduce more interpretable quantum mechanical energy benchmarks that are grounded in the original creation of the GEOM Drugs dataset. Overall, Megalodon explores improvements in molecule design via its architecture and overall generative modeling paradigm. We also explore the similarities and differences between flow matching and diffusion in this specific case of multi-modal molecule design. Although we explore a portion of scaling Megalodon, further improvements in creating meaningful benchmarks are needed going forward as standard validity and molecule stability metrics are already surpassing 96% and 99%, respectively. Such benchmarks are crucial for understanding how to gauge the value of scaling, (Esser et al., 2024).

## 6 REPRODUCIBILITY STATEMENT

We ensure that our data usage, network architecture design, inference sampling, benchmarks and evaluations, and baseline comparisons are reproducible. We point to all necessary prior work for details as our experiments were designed to operate in identical use cases using the same code, including data loaders as prior methods and identical hyperparameters, to best compare our architecture and scaling improvements. Specifically, we use the public repositories for EQGAT-diff and SemlaFlow to run our diffusion and flow matching models with identical set up and hyperparameters. For specific architecture implementations, we point to DiT public repo, EGNN + cross product, and §C for specific implementation details.

## 7 ETHICS STATEMENT

De novo 3D molecule generation is a critical challenge in drug discovery with the potential to revolutionize therapeutic design. The creation of accurate molecular structures is key to unlocking new treatments for a wide range of diseases. While Megalodon offers significant advancements in 3D molecule generation with promising applications, it is important to recognize the potential risks, including biological safety concerns. Generative models for molecular design must be applied responsibly, ensuring their use aligns with ethical standards and safeguards against misuse. Caution is essential when deploying generative models like Megalodon.

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

# Appendix

## Table of Contents

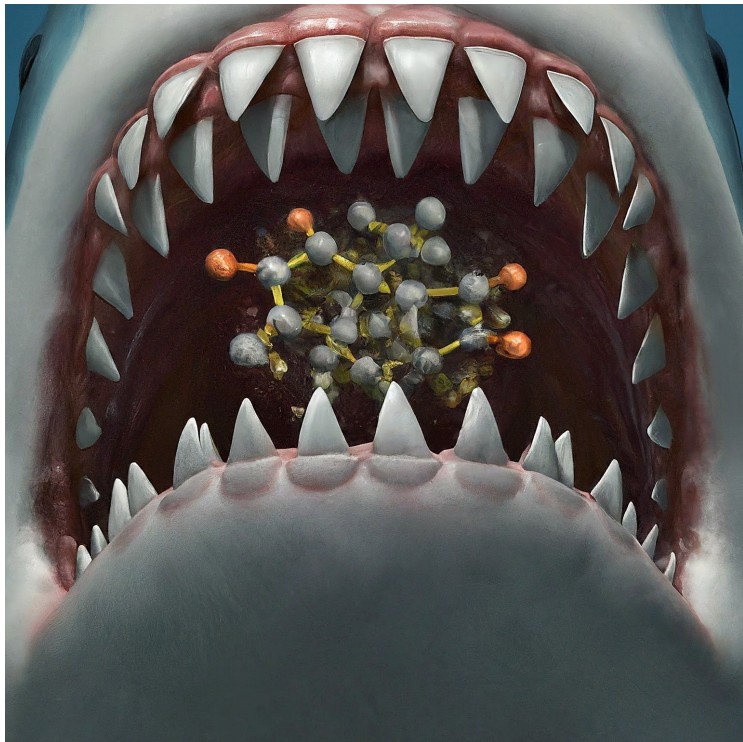

Figure 4: Megalodon molecule generation dynamics generated with Imagen 2

## A  EQUATING CONTINUOUS GAUSSIAN DIFFUSION AND FLOW MATCHING

A part of our work was to explore when to use diffusion versus flow matching and what the empirical differences are. We show below that from a training perspective in the continuous domain, they can be made equivalent.

It can be shown that this objective under the Gaussian setting is a time-dependent scalar multiple of the standard denoising objective explored in Ho et al. (2020). Let's insert equation 1b into the flow matching objective

$$\mathcal{L}_{\text{CFM}}(\theta) = \mathbb{E}_{t, \boldsymbol{\epsilon} \sim \mathcal{N}(\boldsymbol{\epsilon}; \mathbf{0}, \boldsymbol{I}), \mathbf{x}_1 \sim p_{\text{data}}(\mathbf{x}_1)} || \mathbf{v}_\theta(t, \mathbf{x}_t) - \dot{\alpha}(t)\boldsymbol{\epsilon} - \frac{\dot{\beta}(t)}{\beta(t)}(\mathbf{x}_t - \alpha(t)\boldsymbol{\epsilon}) ||^2. \tag{11}$$

where the dot notation denotes the partial time derivative.

Now we see that we can construct an objective that is similar to the "noise prediction" objective that is used in diffusion models:

$$\mathcal{L}_{\text{CFM}}(\theta) = \mathbb{E}_{t, \boldsymbol{\epsilon} \sim \mathcal{N}(\boldsymbol{\epsilon}; \mathbf{0}, \boldsymbol{I}), \mathbf{x}_1 \sim p_{\text{data}}(\mathbf{x}_1)} || \mathbf{v}_\theta(t, \mathbf{x}_t) - \dot{\alpha}(t)\boldsymbol{\epsilon} - \frac{\dot{\beta}(t)}{\beta(t)}(\mathbf{x}_t - \alpha(t)\boldsymbol{\epsilon}) ||^2$$

$$= \mathbb{E}_{t, \boldsymbol{\epsilon} \sim \mathcal{N}(\boldsymbol{\epsilon}; \mathbf{0}, \boldsymbol{I}), \mathbf{x}_1 \sim p_{\text{data}}(\mathbf{x}_1)} || \mathbf{v}_\theta(t, \mathbf{x}_t) - \frac{\dot{\beta}(t)}{\beta(t)}\mathbf{x}_t - \underbrace{(\dot{\alpha}(t) - \frac{\dot{\beta}(t)}{\beta(t)}\alpha(t))}_{=:s(t)} \boldsymbol{\epsilon} ||^2 \tag{12}$$

$$= \mathbb{E}_{t, \boldsymbol{\epsilon} \sim \mathcal{N}(\boldsymbol{\epsilon}; \mathbf{0}, \boldsymbol{I}), \mathbf{x}_1 \sim p_{\text{data}}(\mathbf{x}_1)} s^2(t) || \underbrace{\frac{1}{s(t)}\left( \mathbf{v}_\theta(t, \mathbf{x}_t) - \frac{\dot{\beta}(t)}{\beta(t)}\mathbf{x}_t \right)}_{=:\boldsymbol{\epsilon}_\theta(t, \mathbf{x}_t)} - \boldsymbol{\epsilon} ||^2$$

$$= \mathbb{E}_{t, \boldsymbol{\epsilon} \sim \mathcal{N}(\boldsymbol{\epsilon}; \mathbf{0}, \boldsymbol{I}), \mathbf{x}_1 \sim p_{\text{data}}(\mathbf{x}_1)} s^2(t) || \boldsymbol{\epsilon}_\theta(t, \mathbf{x}_t) - \boldsymbol{\epsilon} ||^2.$$

We see that the resulting mean squared error of noise prediction is the original core loss derived in Ho et al. (2020). This allows us to choose time-dependent scalars via the time distribution itself or the noise or variance schedule to equate the CFM and Diffusion objectives.

In the generative modeling case, we interpolate between a data distribution and a Gaussian density, meaning all data-conditional paths are Gaussian. In that special case, we can, in fact, easily extract the score function from the regular flow matching objective, and we get stochastic sampling for free. We know that $\mathbf{x}_t \sim p(\mathbf{x}_t | \mathbf{x}_1)$ follows Gaussian probability paths. Based on equation 1, we know that

$$\mathbf{x}_t \sim p(\mathbf{x}_t | \mathbf{x}_1) = \mathcal{N}(\mathbf{x}_t; \beta(t)\mathbf{x}_1, \alpha^2(t)\boldsymbol{I}). \tag{13}$$

Let's calculate the score:

$$\nabla_{\mathbf{x}_t} \log p(\mathbf{x}_t | \mathbf{x}_1) = -\nabla_{\mathbf{x}_t} \frac{(\mathbf{x}_t - \beta(t)\mathbf{x}_1)^2}{2\alpha^2(t)}$$

$$= -\frac{\mathbf{x}_t - \beta(t)\mathbf{x}_1}{\alpha^2(t)} \tag{14}$$

$$= -\frac{\boldsymbol{\epsilon}}{\alpha(t)},$$

where we used equation 1 in the last step. We can solve this for $\boldsymbol{\epsilon}$ and insert into the reparametrized $\mathcal{L}_{\text{CFM}}$ in equation 12 and see that we obtain denoising score matching Vincent (2011), which implies that $\boldsymbol{\epsilon}_\theta(t, \mathbf{x}_t)$, or analogously $\mathbf{v}_\theta(t, \mathbf{x}_t)$ via their connection, learn a model of the marginal score $\nabla_{\mathbf{x}_t} \log p(\mathbf{x}_t)$.

Specifically, we have alternatively

$$\boldsymbol{\epsilon}_\theta(t, \mathbf{x}_t) = -\alpha(t)\nabla_{\mathbf{x}_t} \log p(\mathbf{x}_t), \tag{15}$$

$$\mathbf{v}_\theta(t, \mathbf{x}_t) = -\alpha(t)\frac{\beta(t)\dot{\alpha}(t) - \dot{\beta}(t)\alpha(t)}{\beta(t)}\nabla_{\mathbf{x}_t} \log p(\mathbf{x}_t) + \frac{\dot{\beta}(t)}{\beta(t)}\mathbf{x}_t. \tag{16}$$

We note that these equations only hold for a Gaussian prior without optimal transport.

## B    RELATED WORK

Here we discuss other approaches for unconditional molecule generation we find relevant in the context our our study that were not already discussed in Sec. 2. Xu et al. (2023) introduces GeoLDM a geometric latent diffusion model for 3DMG. GeoLDM applies its diffusion process over a learned latent representation. So rather than updating the atom position and types in euclidean space everything is done inside the model. Similar to EDM, GeoLDM uses OpenBabel for bond prediction. Pinheiro et al. (2024) takes a different approach than majority of prior work in representing molecules as 3D voxels rather than graphs. This is akin to 3D image processing rather than point cloud processing. This however requires a recovery process as the voxels are not a natural molecule representation. Voxels however provide a better link to the applications of vision models which majority of the diffusion framework was created for. Lastly, Song et al. (2024) introduces GeoBFN a Geometric Bayesian Flow Network, that unlike diffusion models operate in the parameter space rather then product space. While the integration of 3D voxels would not work for Megalodon , latent diffusion and BFN extensions are something relevant to future work.

## C    MEGALODON ARCHITECTURE

### C.1    ARCHITECTURE

As described in  Fig. 1, Megalodon consists of N augmented transformer blocks that consist of a fused DiT block and a structure layer.

| Parameter | Megalodon Small | Megalodon Large |
|---|---|---|
| Invariant Edge Feature Dimension | 64 | 256 |
| Invariant Node Feature Dimension | 256 | 256 |
| Number of Vector Features | 64 | 128 |
| Number of Layers | 10 | 10 |
| Number of DiT Attention Heads | 4 | 4 |
| Distance Feature Size | 16 | 128 |

Table 4: Comparison of Megalodon Small and Megalodon Large hyperparameter configurations.

#### C.1.1    INPUT/OUTPUT LAYERS

Megalodon takes the input molecules structures and projects them into a $N \times D$ tensor where $D$ is the number of vector features. After all augmented transformer blocks, the predicted structure is projected back down to $N \times 3$.

Similarly, the input discrete components are projected from their one hot variable to a hidden dimension size. The bonds leverage the edge feature size, and the atom types and charges use the node feature size. After all augmented transformer blocks, final prediction heads are applied to project the values back into their respective vocabulary size for discrete prediction.

#### C.1.2    DIT BLOCK

Our DiT block is based on that which was introduced in Peebles & Xie (2022) with a few key differences.

- Rather than just operating over the discrete atom type features $H$, we operate over a fused feature $m = \frac{1}{N} \sum_{i,j \in N} f\left(h_{\mathrm{norm},i,j}, h_{\mathrm{norm},i,j}, \mathrm{e}_{\mathrm{norm},i,j}, \mathrm{distance}_{i,j}\right)$ where $h_{\mathrm{norm}}$ and $e_{\mathrm{norm}}$ are the outputs of the time conditioned adaptive layer norm for the atom type and edge type features. The distance features are the concatenation of scalar distances and dot products. We note that this fusing step is important to ground the simple equivariant structure update layer to the transformer trunk.

- We employ query key normalization (Henry et al., 2020; Hayes et al., 2024).

- The multi-head attention is applied to $m$ to produce mha_out and then used directly in the standard DiT feed-forward to produce $H_{out}$. To create $E_{out}$ we mimic the same steps but use $f(\text{mha\_out}_i + \text{mha\_out}_j)$ for all edges between nodes $i$ and $j$. Our feed-forward is the standard SWiGLU layer with a feature projection of 4. We note that this feed-forward for edge features is the most expensive component of the model, which is why Megalodon-small is designed the way it is.

### C.1.3 Structure Layer

Following Schneuing et al. (2022), the structure layer of Megalodon consists of a single EGNN layer with a positional and cross-product update component. Before this operation, all inputs are normalized to prevent value and gradient explosion, a common problem faced when using EGNNs (Satorras et al., 2021). The invariant features use standard layer norm, whereas the equivariant features use an E3Norm (Vignac et al., 2023).

$$\boldsymbol{x}_i^{l+1} = \boldsymbol{x}_i^l + \sum_{j \neq i} \frac{\boldsymbol{x}_i^l - \boldsymbol{x}_j^l}{d_{ij} + 1} \phi_x^d(\boldsymbol{h}_i^l, \boldsymbol{h}_j^l, d_{ij}^2, a_{ij}) +$$

$$\frac{(\boldsymbol{x}_i^l - \bar{\boldsymbol{x}}^l) \times (\boldsymbol{x}_j^l - \bar{\boldsymbol{x}}^l)}{||(\boldsymbol{x}_i^l - \bar{\boldsymbol{x}}^l) \times (\boldsymbol{x}_j^l - \bar{\boldsymbol{x}}^l)|| + 1} \phi_x^\times(\boldsymbol{h}_i^l, \boldsymbol{h}_j^l, d_{ij}^2, a_{ij}), \quad (17)$$

### C.2 Compute and Data Requirements

Similar to Le et al. (2024), we use MiDi's adaptive dataloader for GEOM DRUGS with a batch cost of 200. We note that the adaptive logic randomly selects one molecule and fills in the batch with similar-sized molecules, tossing any molecules selected that do not fit the adaptive criteria out of the current epoch's available molecules. As a result, an epoch in this setting does not hold the standard connotation as time for the model to see each training data point. We use this dataloader as it was used by prior methods and we felt it important to standardize the data to best create a fair comparison. Megalodon-small is trained on 4 NVIDIA A100 GPUs for 250 epochs. Megalodon was trained on 8 A100 GPUs for 250 epochs, taking roughly 2 days.

Megalodon-flow was trained using the data splits and adaptive data loader from Irwin et al. (2024), which does not discard molecules though was prefiltered to only include molecules with $\leq 72$ atoms. It was trained for 200 epochs on 8 A100 NVIDIA GPUs.

## D Extended Unconditional Generation

### D.1 Unconditional Ablations

We include each primary model in its base form as well as with 5x fewer inference steps. The flow models do not have to be retrained as they were trained to learn a continuous vector field, whereas the diffusion models must be retrained due to the change in variance discretization in the forward diffusion process.

We also include EGNN + cross product which is similar to Megalodon except the transformer layers were replaced by the standard invariant and edge feature updates in Satorras et al. (2021). Prior methods exist that improve upon EDM + Open Babel and maintain that bonds are generated external to the model via Open Babel (Song et al., 2023; Peng et al., 2023). We do not include such methods in our comparison as, for the most part, public code with weights is not available, and Open Babel introduces significant bias and errors, which make evaluating the model difficult (Song et al., 2023; Walters, 2024).

Open Babel, while a powerful tool for molecular manipulation and conversion, can introduce several potential errors, particularly in the context of bond assignment and 3D structure generation. Some common errors include:

- Incorrect bond orders: Open Babel often assigns bond orders based on geometric heuristics or atom types, which can lead to inaccuracies, especially in complex or exotic molecules where bond orders are not trivial.

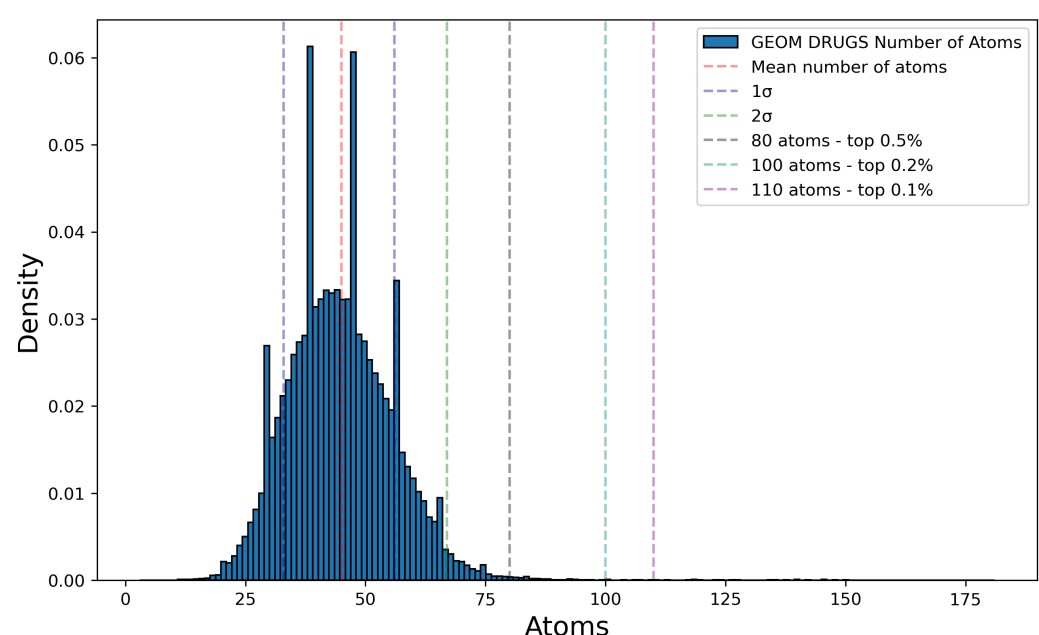

Figure 5: Distribution of molecule sizes

Table 5: Measuring Unconditional Molecule Generation: 2D topological and 3D distributional benchmarks. * Denotes taken from EQGAT-Diff.

| Model | Steps | 2D Topological (↑) | | | 3D Distributional (↓) | |
|---|---|---|---|---|---|---|
| | | Atom Stab. | Mol Stab. | Connected Validity | Bond Angle | Dihedral |
| EDM + OpenBabel* | 1000 | 0.978 | 0.403 | 0.363 | – | – |
| MiDi* | 500 | 0.997 | 0.897 | 0.705 | – | – |
| EQGAT-diff$_{disc}^{x0}$ | 100 | 0.996 | 0.891 | 0.768 | 1.772 | 3.514 |
| EQGAT-diff$_{disc}^{x0}$ | 500 | 0.998 | 0.935 | 0.830 | 0.858 | 2.860 |
| EGNN + cross product | 500 | 0.982 | 0.713 | 0.223 | 14.778 | 17.003 |
| Megalodon-small | 500 | 0.998 | 0.961 | 0.900 | 0.689 | 2.383 |
| Megalodon | 100 | 0.998 | 0.939 | 0.817 | 0.871 | 3.367 |
| Megalodon | 500 | **0.999** | 0.977 | 0.927 | **0.461** | **1.231** |
| SemlaFlow | 20 | 0.997 | 0.962 | 0.875 | 2.188 | 3.173 |
| SemlaFlow | 100 | 0.998 | 0.979 | 0.920 | 1.274 | 1.934 |
| Megalodon-flow | 20 | 0.996 | 0.964 | 0.886 | 1.892 | 3.184 |
| Megalodon-flow | 100 | 0.997 | **0.990** | **0.948** | 0.976 | 2.085 |

- Geometric distortions: When converting between different formats or generating 3D coordinates, Open Babel may generate suboptimal or distorted geometries, especially if the input structure is incomplete or poorly defined.

- Protonation state assumptions: Open Babel may incorrectly infer or standardize protonation states, which can lead to chemical inaccuracies, especially in sensitive systems such as drug-like molecules or biologically active compounds.

- Ambiguous aromaticity: Open Babel can sometimes misinterpret or incorrectly assign aromaticity, which can lead to an incorrect representation of the molecular structure.

- Missing stereochemistry: While converting or generating structures, stereochemistry can be incorrectly assigned or lost altogether, affecting the overall molecular properties.

## D.2 3D DISTRIBUTIONAL METRICS

To evaluate the geometric fidelity of the generated molecules, we compute the Wasserstein-1 distance between the generated and target distributions of bond angles, following the methodology of (Le et al., 2024). The overall bond angle metric is defined as:

$$W_{\text{angles}} = \sum_{y \in \text{atom types}} p(y) \cdot W_1\big(\hat{D}_{\text{angle}}(y), D_{\text{angle}}(y)\big),$$

where $p(y)$ is the probability of atom type $y$, $W_1$ denotes the Wasserstein-1 distance, $\hat{D}_{\text{angle}}(y)$ is the bond angle distribution for atom type $y$ in the generated data, and $D_{\text{angle}}(y)$ is the corresponding distribution in of test set.

Similarly, for torsion angles, the metric is calculated as:

$$W_{\text{torsions}} = \sum_{y \in \text{bond types}} p(y) \cdot W_1\big(\hat{D}_{\text{torsion}}(y), D_{\text{torsion}}(y)\big),$$

where $p(y)$ is the probability of bond type $y$, $\hat{D}_{\text{torsion}}(y)$ is the torsion angle distribution for bond type $y$ in the generated data, and $D_{\text{torsion}}(y)$ is the corresponding distribution in the test set. Since we utilized RDKit to identify torsions, the torsional distribution difference was computed only for valid molecules.

## E GEOMETRIC CONFORMATIONAL ANALYSIS BENCHMARKS

To quantitatively evaluate the fidelity of generated molecular structures relative to their relaxed counterparts, we introduce benchmarks that assess differences in bond lengths, bond angles, and torsion angles. These metrics provide detailed insights into how closely the generated conformations approximate local minima on the potential energy surface.

### E.1 BOND LENGTH DIFFERENCES

For each bond in the molecule, we compute the difference in bond lengths between the initial (generated) and optimized (relaxed) structures. Let $r_{ij}^{\text{init}}$ and $r_{ij}^{\text{opt}}$ denote the distances between atoms $i$ and $j$ in the initial and optimized conformations, respectively. The bond length difference $\Delta r_{ij}$ is calculated as:

$$\Delta r_{ij} = \left| r_{ij}^{\text{init}} - r_{ij}^{\text{opt}} \right|$$

We compute average differences and corresponding frequencies for each possible combination of source atom type, bond type, and target atom type. The final result is the weighted sum of those differences.

### E.2 BOND ANGLE DIFFERENCES

For each bond angle formed by three connected atoms $i$, $j$, and $k$, we calculate the angle difference between the initial and optimized structures. Let $\theta_{ijk}^{\text{init}}$ and $\theta_{ijk}^{\text{opt}}$ represent the bond angles at atom $j$ in the initial and optimized conformations, respectively. The bond angle difference $\Delta\theta_{ijk}$ is given by:

$$\Delta\theta_{ijk} = \min\left( \left| \theta_{ijk}^{\text{init}} - \theta_{ijk}^{\text{opt}} \right|, 180° - \left| \theta_{ijk}^{\text{init}} - \theta_{ijk}^{\text{opt}} \right| \right)$$

As with bond lengths, these differences are grouped based on the types of atoms and bonds involved to calculate the final results.

### E.3 TORSION ANGLE DIFFERENCES

Torsion angles involve four connected atoms $i$, $j$, $k$, and $l$. We compute the difference in torsion angles between the initial and optimized structures using:

$$\Delta\phi_{ijkl} = \min\left(\left|\phi_{ijkl}^{\text{init}} - \phi_{ijkl}^{\text{opt}}\right|, \; 360° - \left|\phi_{ijkl}^{\text{init}} - \phi_{ijkl}^{\text{opt}}\right|\right)$$

where $\phi_{ijkl}^{\text{init}}$ and $\phi_{ijkl}^{\text{opt}}$ are the dihedral angles in the initial and optimized conformations, respectively. This formula accounts for the periodicity of dihedral angles, ensuring the smallest possible difference is used.

By analyzing these statistical measures, we can assess the structural deviations of generated molecules from their relaxed forms. Lower average differences indicate that the generative model produces conformations closer to local energy minima.

As with bond lengths, these differences are grouped based on the types of atoms and bonds involved to calculate the final results.

### E.4 xTB ENERGY BENCHMARK

We also computed the median and mean relaxation energies ($\Delta E_{\text{relax}}$) for both ground truth data and generated molecules using both GFN2-xTB and MMFF force fields. The relaxation energy is defined as the energy difference between the optimized (relaxed) structure and the initial (generated) structure:

$$\Delta E_{\text{relax}} = E_{\text{optimized}} - E_{\text{initial}}$$

#### E.4.1 LIMITATIONS OF MMFF FOR EVALUATING GFN2-xTB STRUCTURES

Previous studies have used the MMFF force field to assess the quality of generated molecular structures (Xu et al., 2022; Irwin et al., 2024). However, the choice of force field is critical when evaluating molecular geometries, as different force fields can yield significantly different energy landscapes. For ground truth conformers optimized using GFN2-xTB, the mean relaxation energy difference $\Delta E_{\text{relax}}$ calculated with GFN2-xTB is nearly zero, as expected. However, when these same structures are evaluated using the MMFF force field, the mean $\Delta E_{\text{relax}}$ is approximately 16 kcal/mol, which aligns with literature values reporting MMFF errors in the range of 15–20 kcal/mol when compared to higher-level methods like GFN2-xTB (Foloppe & Chen, 2019). In other words, xTB is significantly more accurate than MMFF, especially for data generated with xTB.

In contrast, our generated molecules exhibit a mean $\Delta E_{\text{relax}}$ of around 5 kcal/mol when relaxed with GFN2-xTB, significantly smaller than the error observed when using MMFF. This demonstrates that our model produces structures that are much closer to the GFN2-xTB energy minima compared to what MMFF evaluations suggest. These substantial energy discrepancies—stemming from systematic differences like optimal bond lengths and angles—highlight that MMFF is inappropriate for evaluating structures optimized or generated within the GFN2-xTB framework. Relying on MMFF can, therefore, lead to misleading assessments of structural quality.

#### E.4.2 ACHIEVING THERMODYNAMICALLY RELEVANT ENERGY ACCURACY

In statistical thermodynamics, conformers exist in dynamic equilibrium, and their population distribution is determined by their relative free energies. The equilibrium constant $K$ between two conformers is given by:

$$K = e^{-\Delta G^\circ / RT},$$

where $\Delta G^\circ$ is the standard free energy difference, $R$ is the gas constant, and $T$ is the temperature in Kelvin. At room temperature (298 K), the thermal energy $RT$ is approximately 0.6 kcal/mol. A free energy difference of 1.36 kcal/mol corresponds to a tenfold difference in the equilibrium constant.

The GEOM dataset selects conformers within a 2.5 kcal/mol energy window, encompassing about 99.9% of the Boltzmann population for the lowest-energy conformers.

Our generated molecules have a median relaxation energy $\Delta E_{\text{relax}}$ of around 3kcal/mol, approaching this thermally relevant interval. Notably, our model is the first to achieve such proximity to the thermodynamic threshold, establishing a significant milestone in the generative modeling of molecular conformations. By generating conformations with relaxation energies close to the thermal energy interval, our model effectively produces energetically feasible structures near local minima on the GFN2-xTB potential energy surface. This breakthrough demonstrates the model's potential for practical applications, such as conformational searches and drug discovery, where accurate conformer generation is crucial.

By measuring these geometric deviations and appropriate relaxation energy, we offer a comprehensive evaluation of the accuracy of generated molecular conformations, facilitating the development of more precise generative models in computational chemistry.

Table 6: xTB Relaxation Error: Length Å, angles degrees, energy kcal/mol. These metrics are taken over the valid molecules from Table 1. Methods are grouped by model type: diffusion (500 steps) and flow matching (100 steps)

| Model | Bond Length | Bond Angles | Dihedral | Median $\Delta E_{\text{relax}}$ | Mean $\Delta E_{\text{relax}}$ | Mean $\Delta E_{\text{relax}}^{\text{MMFF}}$ |
|---|---|---|---|---|---|---|
| GEOM-Drugs | 0.0 | 0.0 | 7.2e-3 | 0.00 | 1.0e-3 | 16.48 |
| EQGAT-diff | 0.0076 | 0.95 | 7.98 | 6.36 | 11.06 | 28.45 |
| Megalodon-small | 0.0085 | 0.88 | 7.28 | 5.78 | 9.74 | 24.87 |
| Megalodon | **0.0061** | **0.66** | **5.42** | **3.17** | **5.71** | 21.61 |
| SemlaFlow | 0.0309 | 2.03 | 6.01 | 32.96 | 93.13 | 69.46 |
| Megalodon-flow | **0.0225** | **1.59** | **5.49** | **20.86** | **46.86** | 45.51 |

## F    MEGALODON MOLECULE VISUALIZATION

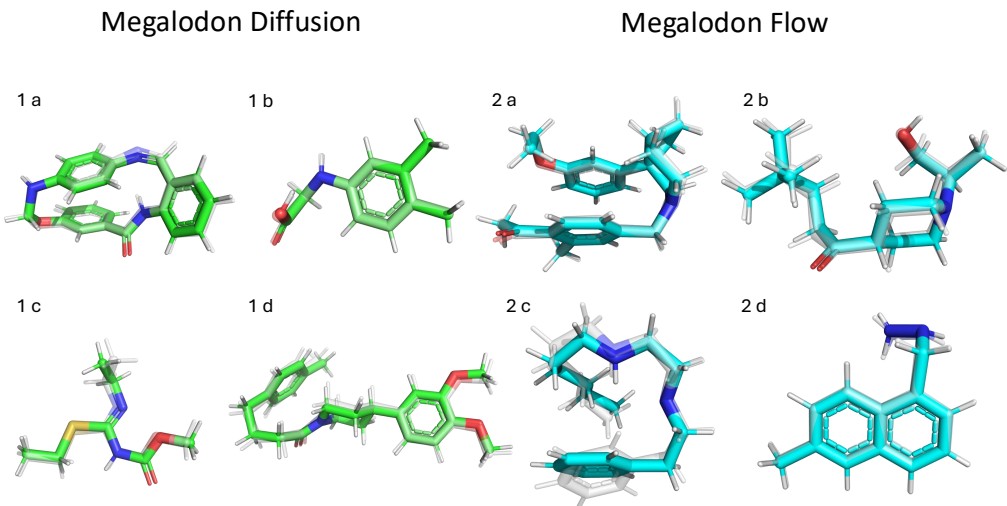

Figure 6: Examples of generated molecules using Megalodon: (1) Diffusion and (2) Flow Matching. Each generated molecule is displayed alongside its corresponding optimized structure (shown in transparent grey). The examples include small aromatic molecules (1b, 2d), molecules exhibiting pi-stacking interactions (1a, 2a), non-aromatic molecules (1c, 2b), and a molecule with a macrocycle (1a).

## G  LIMITATIONS

While we show that Megalodon performs well across a variety of 3D de novo molecules tasks there are still some limitations that are worthy of discussion.

- Megalodon like  Le et al. (2024) and the prior edge prediction generative models before it relies on maintaining $N^2$ edge features, which is quite expensive. Recently (Irwin et al., 2024) was able to avoid this issue for a majority of the model architecture by fusing the edge and atom features, but this creates a trade-off between model speed and accuracy. Our ablations show that the larger edge features are critical for strong energy performance, so it is still an open question for how to best deal with discrete edge types as each atom can have a maximum of 6 bonds at a time, so is needing to model all $N$ potential pairings at all times really necessary? We leave future work to explore this in greater depth.

- As discussed herein, the existing 3D molecule generation benchmarks are quite limited. A common theme that has been discussed in prior work (Le et al., 2024; Irwin et al., 2024). While we make strides in expanding the field of view of de novo design and energy-based benchmarks. More work needs to be done to measure important qualities, as even for common conditional design benchmarks, metrics such as QED are not meaningful in practice, and even more complex properties like protein-ligand binding affinity can be directly optimized for with non-3D structure-based methods (Reidenbach, 2024). For these reasons, we looked to explore conditional structure generation, but across the board, small molecule benchmarking is a current field-wide limitation when compared to the current drug discovery practices.

- A general limitation for all prior 3DMG models including Megalodon is that they cannot generalize to unseen atom types due to the one hot representation during training. As a result, these models can only be used for GEOM-Drug-like molecules. While fairly extensive, it is a limitation worth noting as the flexibility is limited when compared to 2D or SMILES-based LLMs.

