# OpenReview forum: "Applications of Modular Co-Design  for De Novo 3D Molecule Generation"
_ICLR.cc/2025/Conference — Submitted to ICLR 2025_

### Official Review · Reviewer_nciS · 2024-11-03

**Soundness:** 3
**Presentation:** 3
**Contribution:** 3
**Rating:** 6
**Confidence:** 4

**Summary:**

This paper proposes Megalodon, a transformer-based model for 3D molecule generation. Megalodon is a modular approach with both diffusion and flow matching objectives that aim to improve 3D structure prediction and validity. The authors conducted experiments on existing benchmarks such as GEOM Drugs and introduced new metrics such as xTB relaxation error. The results indicate Megalodon outperforms existing methods in molecule stability, validity, and energy.

**Strengths:**

1. The proposed Megalodon is an adaptive architecture that can be adapted with both diffusion and flow matching objectives.

2. The newly introduced benchmarks including energy-based and 3D structure-based assessments are more aligned with practical applications in molecular design and drug discovery.

3. The experimental results are promising, especially along the metrics related to 3D structures.

**Weaknesses:**

My major concern is the limited technical novelty. The network architecture of Megalodon uses standard DiT models, with only minor modifications such as the structure layer. While the authors introduce a combination of diffusion and flow matching objectives, this integration alone does not constitute a major advancement, as flow matching is a theoretically more general framework than diffusion. There is no surprise that these two objectives can be used in one framework.

**Questions:**

See weakness.

---

> ### Author Response · Authors · 2024-11-16
> **Response**
>
> We thank the reviewer for finding our work adaptable leading to promising results especially for 3D structures. We also appreciate that our new benchmarks were found to be more aligned with practical applications of molecule design and drug discovery.
>
> We thank the reviewer for their time and understand that novelty of a method can be highly debatable. For this reason we will highlight aspects of our work that we find novel below. **If there are more specific questions we are happy to answer them**
>
> # **Architecture Novelty**
>
> The standard DiT block takes in a singular input tensor, H. To enable the modeling of 3D molecules composed of several continuous and discrete data modalities, we made several critical changes to the architecture, which we refer to as fused-DiT (f-DiT).
>  - As described in Sec. B.1.2, our fused DiT blocks take in (X, H, E, C) for the 3D structure and discrete atom, bond, and charge types. These features are first fused and aggregated in a message-passing-like operation. The multi-head self-attention is then applied to these fused features. We note that if the attention is applied to the respective inputs as done in the standard DiT, the model does not converge and outputs 100% invalid molecules with unrealistic structures
> - For the feed-forward part of the f-Dit block, the processed fused features are aggregated along all pairs of nodes to create new bond features, and they are hit with a linear layer to create new atom and charge features. From here, we apply independent feedforward and adaptive layernorm operations for each data modality (excluding structure since the structure is only used as an input to the f-DiT block) to create updated features for all discrete data modalities.
>     - In short H’, E’, C’ = f-DIT(X,H,E,C)
>     - We emphasize that the only operation maintaining equivariance and updating the structure is the series of EGNN single layers. In comparison, **if we replace the f-DiT operation with standard EGNN non equivariant feature updates, we see a drop in Validity of almost 70%, as shown in Table 1**. From our understanding, this is the first work to obtain such strong results with a simple EGNN-based architecture since EDM + Openbabel.
>
> # **Technical Novelty**
>
> Our novelty is further rooted in our applications, evaluations, and analysis, which include the introduction of new benchmarks and the reintroduction of a conditional structure generation task.**
>  - Figure 3 introduces molecule size as a new component to unconditional benchmarking in which the Megalodon significantly outperforms EQGAT-diff.  We demonstrate this performance can be improved with further scaling of our architecture. **The ability to generate 49x more valid and stable molecules compared to prior SOTA** is a significant result, given that both are trained with identical data and diffusion parameterizations.
>  - **Table 2 demonstrates that off-the-shelf 3D molecule generation models cannot be used for conformer generation. In contrast, Megalodon can and surpasses strong conformer baselines due to the use of a compounded time-dependent noise scheduler** described in line 297. This is a novel and quite surprising finding, as we expected all unconditional molecule generation models to be able to conditionally generate structure as they are trained with independent structure and discrete denoising.
> - Furthermore, when compared to GeoDiff, which also uses an EGNN-based architecture with identical diffusion parameterization, **we demonstrate that unconditional generative pretraining is extremely beneficial in generating better structures in 10x fewer sampling steps**. In other words, learning how to generate the 2D discrete components improves the ability to generate accurate conformers.
> - We demonstrate that with the **diffusion** objective, Megalodon is capable of generating conformers—that is, molecules very close to their local minima of the ground truth energy function. The median relaxation energy drop of **3.17 kcal/mol** approaches the threshold of **2.5 kcal/mol**, which is often considered the thermodynamically relevant interval. **Furthermore this is 2-10x better than prior methods**. This proximity emphasizes the potential practical value of our method. In contrast, we showed that with the **Flow Matching** objective, the energy drop is an **order of magnitude larger**, highlighting a significant and valuable difference for readers.
> - **Subsequently, we uncover an efficiency and accuracy tradeoff between FM and DM for 3DMG**. FM yields more valid models and can be constrained to very few sampling steps, whereas DM exhibits an order of magnitude better structure measured by the molecular energy.
>
> **Overall**,  we are the first to perform a comprehensive analysis of the interplay between the 2D graph and 3D structure during molecular generation in previous methods and generative frameworks(Diffusion vs Flow Matching) and offer potential solutions to improve the dependency between the modalities.

---

> > ### Comment · Reviewer_nciS · 2024-11-25
> >
> > Thanks for the authors' response. I am convinced the empirical performance is significant and I acknowledge the authors' claims on novelties. However, I cannot champion the paper since I still think the technical modifications are rather minor. I will keep my current positive score.

---

### Official Review · Reviewer_PoXh · 2024-11-03

**Soundness:** 4
**Presentation:** 4
**Contribution:** 3
**Rating:** 8
**Confidence:** 4

**Summary:**

This work presents a transformer-based diffusion and flow matching framework for the co-design of 2D and 3D molecular structures. The coordinate, atom and bond features of the noisy molecule are aggregated through DiT blocks and then used to reconstruct the 3D and 2D structures with an EGNN layer. The authors establish the model architecture for both diffusion and flow matching. The proposed model shows higher generation quality in both manners, especially for larger molecules.

**Strengths:**

1. The proposed model shows overall higher performance, especially on the more challenging task of generating larger molecules, while also having better memory efficiency than the previous models.
2. The authors perform comprehensive analysis of the interplay between the 2D graph and 3D structure during molecular generation in previous methods, and offer potential solutions to improve the dependency between the modalities.
3. This paper also attempts to build a framework adaptable to multiple training methods (diffusion and flow matching), which would be informative for future studies.
4. The authors also offer additional benchmark tasks and metrics for evaluating 3D molecule generation.

**Weaknesses:**

See Questions.

**Questions:**

1. How is equivariance preserved? From Appendix B.1.3, it seems the structure blocks should also take the input coordinates and combine them with the DiT block output to update the structure. Intuitively, there should be a skip connection from the input 3D coordinates to the structure blocks. Otherwise the coordinate information would be lost. However, Fig 1 indicates the DiT blocks and structure layers are sequential, where the structure blocks only take the DiT output (which is invariant) to predict the structure. Could the authors clarify on this?
2. For conditional generation, how is the 2D graph information provided to the diffusion model?

---

> ### Author Response · Authors · 2024-11-16
> **Response**
>
> We thank the reviewer for highlighting key contributions of Megalodon, including its ability to advance 3D molecule generation (3DMG) into practical applications with improved performance on larger molecules and practical structural benchmarks. We’re glad that our in-depth analysis of the 2D-3D interplay in molecular generation was found to be thorough and valuable.
>
>
> We are happy to provide more details on the provided questions below.
>
> # **Q1. How is equivariance preserved?**
>
> Thank you for pointing out the inconsistency in Figure 1, and we apologize for any confusion this may have caused. There is a skip connection seen in Eqn. 17, which we will add to the camera-ready figure. In our architecture, the DiT (Diffusion Transformer) block operates on invariant features such as atom types, charges, bond types, pairwise atom distances, and coordinate norms. The output of the DiT block consists of an updated atom (H), charge (C), and bond (E). In practice, C is concatenated to H but we write it explicitly here to be more clear.
>
> - H’, E’, C’ = f-DIT(X,H,E,C)
> - X’ = EGNN_X_Only(X, H’, E’, C’) #only x update see Eqn. 17
>
> The structure block then uses these updated atom and bond features to update the coordinates. Since we subtract the center of mass (CoM) from the coordinates, all features processed by the DiT block are invariant under rotations and translations. Therefore, the DiT block updates only invariant features.
>
> Our structure block follows the Equivariant Graph Neural Network (EGNN) architecture [Satorras et al., 2021 https://arxiv.org/pdf/2102.09844], as illustrated in the formula in Figure 1. This design ensures that the overall architecture is equivariant, meaning that any rotations or translations applied to the input coordinates result in corresponding rotations or translations in the output coordinates. The skip connections from the input 3D coordinates to the structure block are implicit in the EGNN framework.
>
> # **Q2. How is the 2D graph provided to the diffusion model?**
>
> To provide the **2D graph** to the diffusion model, we supply the atom types (**H**), bond types (**E**), and charge types (**C**) as fixed inputs, while the coordinates (**X**) are generated by the model.
>
> In standard diffusion and flow matching models that directly operate on bonds, the initial inputs H,E,C,X are generated from a prior distribution. At each diffusion step t, the model takes the noised versions of these variables Ht,Et,Ct,Xt​ and predicts the ground truth values.
>
> To enable conditional generation, we modified the training process so that for a fraction of the time the model is conditioned on the ground truth H,E,C, which are supplied to the model as one-hot vectors, and we use RDKit to compute adjacency matrix, bond orders, atom types and formal charges.  This means that during conditional generation, the model receives the fixed 2D graph (represented by H,E,C) and the noised coordinates Xt​, and it predicts the denoised coordinates X.
>
> In summary, the model takes the fixed atom types, bond types, and charges as inputs and generates the corresponding 3D coordinates, effectively performing conditional generation based on the provided 2D molecular graph.
>
>
> ## **If you have any future concerns or questions, we will be happy to address them.**

---

> > ### Comment · Reviewer_PoXh · 2024-11-21
> >
> > Many thanks to the authors for addressing my questions and providing more details.

---

### Official Review · Reviewer_dknQ · 2024-11-04

**Soundness:** 3
**Presentation:** 2
**Contribution:** 1
**Rating:** 5
**Confidence:** 4

**Summary:**

In this paper, the authors propose a method for unconditional 3D molecule generation. The proposed approach, called Megalodon, represents molecules with both 3D structure and 2D topology information (atom coordinates and types, bond types and formal charge). The model uses a transformer-based architecture and either diffusion or flow-matching generative model. The proposed approach achieves positive results on experiments on GEOM-drugs dataset.

**Strengths:**

- The task of molecule generation is important and worth investigating (although the utility of _unconditional_ generation can be discussed)
- The paper achieved good experimental results on GEOM-drugs dataset.
- The paper shows that having a better architecture (ie transformer) and more parameters can help on GEOM-drugs molecule generation.

**Weaknesses:**

- The paper is not very well written and could be improved. In particular, there is a lot of training/evaluation details missing, making reproducibility challenging.
- The paper lacks novelty. The paper uses well-stablished generative models (diffusion or flow matching, already used many times on this task) on a single standard dataset (GEOM-drugs).
- Many of the parameters choices were made ad hoc. It would be great to see some ablation studies to justify many of the architecture choices made by the authors (eg, the self conditioning, the modifications on DiT architecture).
- The authors only show results in one single dataset (GEOM-drugs). It would be nice to see results in other datasets to make sure results are generalizable, eg QM9, PubChem3D, or other related tasks that relies on different dataset (eg, structure-condition generation instead of only conformer generation), etc.
- The paper misses a lot of references/comparison to related works: eg, GeoLDM (Xu et al, ICML23), VoxMol (Pinheiro et al, NeurIPS23), GeoBFN (Song et al, ICLR24). All these works also explore the problem of unconditional molecule generation. Moreover, the authors wrongly cite MolDiff (Xu et al 23), mentioning that they dont model bond, while they actually do (L120).

**Questions:**

- Why use only a subset of the metrics proposed by the MiDi paper on Table 1, instead of all the metrics? Also, why ignore the "3D distributional" metrics from MiDi?
- Could the authors elaborate more on how the "self-conditioning" is applied? WHy the choice of using it vs not using it? What is the contribution of self conditioning?
-  DiT is a model created to operate on images and much of its inductive bias operate on that data domain. Why did the authors decide to use DiT on their model? What about any other transformer-like architecture? DiT also has a autoencoder to go from pixel to latent space, and it seems that the proposed model does not have that.
- From my understanding, the architecture is composed of equivariant and non-equivariant layers (which end up being a non-equivarant model). Is this correct? If so, why this design choice?
- Could the authors elaborate on why did the proposed model is able to generate conformations from molecular graph, while EQGAT-Diff does not? From my understanding the only difference between the two models is the neural network architecture, and it seems quite surprising that this makes such a difference on teh results.

---

> ### Author Response · Authors · 2024-11-16
> **Response**
>
> Thank you very much for your effort in reviewing our paper and your feedback. We appreciate the depth of the review and give our best effort to answer all questions.
>
> # **W1: Missing training and evaluation details missing**
>
> We provide links to the corresponding code bases from which we used identical training setups for the diffusion and flow matching models. We do this to create a true 1:1 comparison with both EQGAT-Diff and SemlaFlow, with the only change being the architecture we describe in Appendix Section B, which includes all key equations and hyperparameters. This allows us to use the same data loaders and generative model setup as each method, which is important as SemlaFlow filters out all molecules with greater than 75 atoms. We also provide all evaluation details and use standard DDPM SDE and flow matching ODE sampling as done in prior work.
>
> Outside of the training and evaluation details that we clarified above, are there any sections of the paper that were not clear?
>
> # **W2-A: Architecture Novelty**
>
> Our work is more than applying diffusion (DM) and flow matching (FM) for unconditional molecule generation.
>
> First, we want to clarify that DiT Block, while used for images in the original paper, does not have any specific inductive bias for images. As discussed here https://www.wpeebles.com/DiT, the DiT block is just a standard transformer block with an adapted layer norm to handle the conditioning input. It was used for latent image diffusion, but that use case has no bearing on how we augment it for molecule generation.
>
> We emphasize that we deconstructed molecule generation into simultaneous equivariant structure prediction and non-equivariant discrete data prediction (atom, bond, and charge types). This is reflected in our architecture design, which uses an augmented DiT to model the discrete data and then simple EGNN layers to update the structure. Figure 1 illustrates how the model comprises N segments of a DiT block followed by a simple structure update. We chose this architecture to enable better modeling of discrete data via the multi-head self-attention as the discrete data accuracy determines molecule stability and validity.
>
> The standard DiT block takes in a singular input tensor, H. To enable the modeling of 3D molecules composed of several continuous and discrete data modalities, we made several critical changes to the architecture, which we refer to as fused-DiT (f-DiT).
>  - As described in Sec. B.1.2, our fused DiT blocks take in (X, H, E, C) for the 3D structure and discrete atom, bond, and charge types. These features are first fused and aggregated in a message-passing-like operation. The multi-head self-attention is then applied to these fused features. We note that if the attention is applied to the respective inputs as done in the standard DiT, the model does not converge and outputs 100% invalid molecules with unrealistic structures.
> - For the feed-forward part of the f-Dit block, the processed fused features are aggregated along all pairs of nodes to create new bond features, and they are hit with a linear layer to create new atom and charge features. From here, we apply independent feedforward and adaptive layernorm operations for each data modality (excluding structure since the structure is only used as an input to the f-DiT block) to create updated features for all discrete data modalities.
>     - In short H’, E’, C’ = f-DIT(X,H,E,C)
>     - We emphasize that the only operation maintaining equivariance and updating the structure is the series of EGNN single layers. In comparison, **if we replace the f-DiT operation with standard EGNN non equivariant feature updates, we see a drop in Validity of almost 70%, as shown in Table 1**. From our understanding, this is the first work to obtain such strong results with a simple EGNN-based architecture since EDM + Openbabel.

---

> > ### Author Response · Authors · 2024-11-16
> > **Response-2**
> >
> > # **W2-B: Lacks Novelty**
> >
> > **Our novelty is further rooted in our applications, evaluations, and analysis, which include the introduction of new benchmarks and the reintroduction of a conditional structure generation task.**
> >  - Figure 3 introduces molecule size as a new component to unconditional benchmarking in which the Megalodon significantly outperforms EQGAT-diff.  We demonstrate this performance can be improved with further scaling of our architecture. **The ability to generate 49x more valid and stable molecules compared to prior SOTA** is a significant result, especially given that both are trained with identical data and diffusion parameterizations.
> >  - **Table 2 demonstrates that off-the-shelf 3D molecule generation models cannot be used for conformer generation. In contrast, Megalodon can and surpasses strong conformer baselines due to the use of a compounded time-dependent noise scheduler** described in line 297. This is a novel and quite surprising finding, as we expected all unconditional molecule generation models to be able to conditionally generate structure as they are trained with independent structure and discrete denoising.
> > - Furthermore, when compared to GeoDiff, which also uses an EGNN-based architecture with identical diffusion parameterization, **we demonstrate that unconditional generative pretraining is extremely beneficial in generating better structures in 10x fewer sampling steps**. In other words, learning how to generate the 2D discrete components improves the ability to generate accurate conformers.
> > - We demonstrate that with the **diffusion** objective, Megalodon is capable of generating conformers—that is, molecules very close to their local minima of the ground truth energy function. The median relaxation energy drop of **3.17 kcal/mol** approaches the threshold of **2.5 kcal/mol**, which is often considered the thermodynamically relevant interval. **Furthermore this is 2-10x better than prior methods**. This proximity emphasizes the potential practical value of our method. In contrast, we showed that with the **Flow Matching** objective, the energy drop is an **order of magnitude larger**, highlighting a significant and valuable difference for readers.
> > - **Subsequently, we uncover an efficiency and accuracy tradeoff between FM and DM for 3DMG**. FM yields more valid models and can be constrained to very few sampling steps, whereas DM exhibits an order of magnitude better structure measured by the molecular energy.
> > - We further show step size ablations in Table 5, demonstrating that our model continuously outperforms baselines at reduced step sizes.
> >
> > **Overall**, we are the first to perform a comprehensive analysis of the interplay between the 2D graph and 3D structure during molecular generation, as well as the choice in generative frameworks(Diffusion vs Flow Matching) via interpretable and informative  energy benchmarks.
> >
> > # **W3: parameter choices are made ad hoc**
> >
> > As discussed in line 304, self-conditioning has been heavily explored in several generative models, including prior molecule generation methods like SemlaFlow. Given this, our base model includes self-conditioning, which we define in Equation 9.
> >
> > As for the impact of self-conditioning, it is very subtle, accounting for roughly 1% performance boosts for molecule stability and validity. Both of these metrics, without self-conditioning, still outperform all prior methods. We conduct all further experiments, including conformer generation and the structure-energy benchmarks, with the models trained with self-conditioning as done in SemlaFlow, given they perform better and net no increase in inference cost.
> >
> > As for the choices with the f-DiT architecture, without the specific fusing as discussed in Appendix B.1.2 the model cannot converge and cannot generate any molecules. Outside of the fusing operation, the only change from the standard DiT block is to create pairwise bond features and have parallel and independent feed-forward updates for bond and atom-type features. We stress that the architecture is very similar to the original DiT. Adaptations were required to work on multiple data types simultaneously, and these choices cannot be easily ablated without changing the task definition of a 3D molecule (i.e. removing bonds or atom type prediction). We emphasize that when we removed the fusing operation or the independent bond features updates, the model broke.
> >
> >
> > For these reasons, we provide architecture ablations specifically with DiT vs EGNN seen in Table 1, which shows a 70% drop in validity when DiT is replaced.
> > All other hyperparameters were selected based on prior work and neither model size or any hyperparameters were specifically optimized for. The differences between Megalodon small and large stem from choices to reduce the parameter size and were not chosen based on any benchmarks.

---

> > > ### Author Response · Authors · 2024-11-16
> > > **Response-3**
> > >
> > > # **W4: Single Dataset**
> > >
> > > We agree that structure-conditioned generation is interesting future work and something we can fine-tune the existing Megalodon models in the future.
> > >
> > > One of our primary research focuses was to deeply understand the challenges of modeling the GEOM-Drugs dataset, which has been proven to be significantly more difficult than QM9 in several prior conformer and molecule generation tasks discussed in Sec. 2.3. For these reasons, we chose to exclude QM9 and concentrate on the more complex and realistic GEOM-Drugs dataset to push the boundaries of current methodologies.
> > >
> > > Regarding PubChem3D, although it is an extensive resource, its conformers are generated using OpenEye OMEGA, which rapidly produces 3D structures (approximately 0.1 seconds per conformer on a single core compared to GEOM’s CREST 90 core hours) through algorithmic methods. However, these conformers are not necessarily at energy minima and may not represent stable forms. Since we focus on generating conformers and estimating their stability—by measuring changes when relaxing to the closest local minima—the PubChem3D dataset is less suitable for our purposes than the GEOM generation procedure. Consequently, the structures in PubChem3D are less accurate for our objectives and would not provide meaningful energy measurements for our analyses.
> > >
> > > Given that our work introduces chemically grounded and interpretable structural benchmarks, we plan to extend our approach to applicable datasets beyond GEOM in the future. This will further validate the robustness and generalizability of our models across diverse molecular datasets. Overall, this work provides a strong initial understanding of what models can and cannot do well.
> > >
> > > # **W5: Missing related Work**
> > >
> > > We thank the reviewer for bringing this to our attention. We have adapted our related work to discuss these methods as we agree they are valuable in unconditional molecule generation.
> > >
> > > We have fixed our typo in citing MolDiff and have included comparisons below from the values taken from their paper. We also note these comparisons are not directly 1:1 as MolDiff removed five element types. MolDiff also does not report stability for their method with hydrogens. We focus on generation with explicit hydrogens as done in prior baselines.
> > >
> > > | Metric              | MolDiff | Megalodon FM | Megalodon Large |
> > > |---------------------|---------|--------------|------------------|
> > > | Connected validity  | 0.739   | 0.948        | 0.927           |

---

> > > > ### Author Response · Authors · 2024-11-16
> > > > **Response to Questions**
> > > >
> > > > # **Q1: Subset of MiDi metrics**
> > > >
> > > > We want to clarify that we are using 3D distributional metrics for bond angles and torsion angles in our evaluation. In our description, we cited EQGAT-Diff because we used the implementation of these metrics from the EQGAT-Diff repository. However, these metrics are exactly the same as those used in the MiDi paper and several other studies. As for MiDi specifically, we combined connected and valid as there is little practical value in generating undesired molecule fragments when all training data is connected. **We also report diversity, novelty, and uniqueness in line 365** outside of Table 1, given that all of the values for all methods are so close.
> > > >
> > > > We decided not to include the 3D distributional metric for bond lengths. Although Megalodon performs better on this metric, we found it was not particularly informative. The metric yielded values ranging from 0.0015 for Megalodon Large to 0.0042 for Semla Flow. Even though Megalodon outperforms Semla Flow, EQGAT-Diff, and MiDi on this data, it is challenging to interpret what these numbers actually signify. To create a more interpretable comparison, **Table 3** shows that the average bond length difference between the initially generated structures and the GFN2-xTB relaxed structures is around **0.01 Å.** This provides a much clearer understanding of the metric's scale and highlights the significance of the changes in bond lengths. By focusing on this metric, we can better assess the practical implications of bond length variations in our models.
> > > >
> > > > | Metric                       | Megalodon FM | Semla FM | Megalodon Large | Megalodon Small | EQGAT-Diff   |
> > > > |------------------------------|--------------|----------|----------|----------|---------|
> > > > | Bond Length Distributional  | 0.002804     | 0.004164 | 0.001510 | 0.004018 | 0.003955 |
> > > >
> > > > Moreover, during our previous experiments, we observed that the implementation of these metrics (3D distributional)—as used by MiDi, EQGAT-Diff, and ourselves—is not robust to outlier molecules. Specifically, a single molecule with a poor 3D structure can significantly affect the results due to how the joint support is defined when computing the cumulative distribution function (CDF) before calculating the Wasserstein distance. Because 3DMG models are on the lever where they are able to learn bond distances at a pretty good level (the Wasserstein distance between distributions is on the level of 0.003), this metric is the most sensible to these outliers.
> > > >
> > > > **One of our main goals in the paper was to develop informative and interpretable 3D benchmarks that highlight the value of our results for computational chemists**. Therefore, we focused on metrics that we found most meaningful and did not include every previously coarse metric.
> > > >
> > > > # **Q2: Self-Conditioning what is it and why?**
> > > >
> > > > Self-conditioning (SC) was introduced by Chen et al. (https://arxiv.org/abs/2208.04202) as a way to condition the denoising network on its predictions to further refine them. Since then, SC has been used in several generative models, including prior molecule generation methods and many image and protein generation models, as discussed in line 304.
> > > >
> > > > Following the procedure described in Chen et al., SC during training follows Eqn. 9, in which 50% of the batches first undergo unconditional denoising x_sc = model(x_t). We then augment x_t = f(x_sc, xt) + x_t (where f is a simple MLP), a new design choice to add a residual connection between the model input and self-conditioned output. Finally, we obtain the final prediction, which is x_pred = model(x_t). In the other 50% of batches, we jump straight to x_pred, ignoring x_sc and the augmentation.
> > > > The choice of using SC is a hyperparameter. This is analogous to the number of recycling steps in AlphaFold2.
> > > >
> > > > In our experience, SC helps bump results by ~1% for validity and stability. Given that it is easy to implement and adds no inference cost, we use it as part of the main model that is in line with prior work.
> > > >
> > > > # **Q3: Why DiT**
> > > >
> > > > We sincerely apologize for any confusion we may have caused by not distinguishing DiT from the DiT block. We have updated our paper to clarify that we refer to the DiT block, not the entire DiT model. The entire latent image DiT model does have an autoencoder, but we only use the underlying DiT block here.
> > > >
> > > > As discussed in our response to **W2-A**, the DiT-block itself is just a standard transformer with an adaptive layer norm. We chose it to integrate a standard transformer with the conditional time tensor so that our prediction can be conditioned on the time step. DiT shares the same inductive bias as traditional multi-head self-attention and feed-forward blocks.

---

> > > > > ### Author Response · Authors · 2024-11-16
> > > > > **Response to Questions - part two**
> > > > >
> > > > > # **Q4: Equivariance**
> > > > >
> > > > > Theoretically, our model is no different than the original EGNN. We take in equivariant features (structure) and invariant features (atom types). We only care for equivariance to be preserved for the equivariant features which is done with the single EGNN structure updates.
> > > > >
> > > > > The invariant DiT blocks only update the invariant features analogous to replacing Eqn 6 of Satorras et al. (https://arxiv.org/pdf/2102.09844) use of a standard MLP with a DiT block. Furthermore, our model follows the same proof in Appendix A since we use identical structure updates, and the parameterization of the invariant feature update has no bearing.
> > > > >
> > > > > We designed our architecture this way since molecule generation is very sensitive to the accuracy of the discrete data components which transformer models excel at. If one bond is off or the atom type is wrong, validity and connectivity are broken. Our architecture allows most of the focus to be on learning a good molecule representation and then integrating it in lightweight structure updates.
> > > > >
> > > > > **We stress that our model is equivariant as only equivariant updates directly update the structure data**, and we ensure zero center of mass to prevent bias in the translations following prior work.
> > > > >
> > > > > # **Q5: Why can’t EQGAT-Diff generate conformers?**
> > > > >
> > > > > We were quite surprised that EQGAT-diff could not generate realistic conformers when prompted with the true 2D molecule graph (atom, bond, and charge types). We found that this was because, for the majority of the sampling trajectory, EQGAT generated no bonds and random atom types until the structure prediction started to converge. Although each data modality was being denoised and optimized independently, the was still a learned dependence even with the use of data-like priors.
> > > > >
> > > > > **We discuss in line 294 our solution in which we introduce a change in the training procedure that allows the explicit decoupling of the discrete and continuous learning objectives** by creating an independent time variable while maintaining identical diffusion variance schedules. This is what allows Megalodon to generate conformers, unlike prior 3DMG models.
> > > > >
> > > > > ## **Thank you again for engaging in our work! If you have any further concerns or questions, we will be happy to address them.**

---

> ### Comment · Reviewer_dknQ · 2024-11-28
>
> I appreciate the authors' rebuttal and the clarifications. I also acknowledge the good empirical results on GEOM-drugs. Although this does not necessarily come as much of a surprise given it has more parameters than the compared baselines. I still find the paper lacks novel technical contributions: the architecture (modulo minor modifications), the data representation, and the generative models utilized have already been used in many similar applications. For these reasons, I keep my current rating.

---

> ### Author Response · Authors · 2024-11-28
>
> We thank you for your feedback. While we disagree that our work lacks novelty due to enabling multi variable message passing into the DiT block which was not a minor modification, high quality unconditional generation and high quality conformer generation in and same model for the first time, and also introducing new interpretable benchmarks, we thank you for asking us questions and making our work stronger.
>
> As for diffusion and flow matching we know they have been used before but we are the first to study them together using the same architecture and biological task at the same time. We find our findings of a physical accuracy vs efficiency trade off between diffusion and fm models quite interesting and something potentially meaningful for other applications of these frameworks outside of small molecule design.
>
> Overall we have answered all questions and addressed all concerns that can be addressed in the rebuttal period. We understand novelty is highly debated and value your feedback.

---

### Meta-Review · Area_Chair_qnBB · 2024-12-22

**Metareview:**

The paper proposes a molecular generation framework that utilizes diffusion and flow matching on transformers to generate molecular structures. The empirical results presented focus on the GEOM-drugs dataset, where the model reportedly outperforms existing baselines.

***Strengths:***
- The proposed model shows superior performance in generating larger molecules and demonstrates better memory efficiency compared to prior models.
- The paper provides an analysis of the interaction between 2D graph and 3D structure in molecular generation, proposing improvements on existing methods.
- The framework is adaptable to diffusion and flow matching, which could be beneficial for future research in the field.
- Introduction of additional benchmarks and metrics for 3D molecule evaluation.

***Weaknesses:***
- The technical novelty is limited as the architecture largely employs established models with minor modifications.
- The study's findings are based solely on the GEOM-drugs dataset, limiting the generalizability of the results.
- The paper suffers from writing issue and lacks sufficient training and evaluation details.
- Lack of comparisons with relevant recent works.

We appreciate the authors' efforts in submitting their rebuttal and providing additional explanations. Some of the issues, e.g. writing clarity, is alleviated. However there are two major concerns are not addressed well form the further replies of the reviewers:
- The architecture and method used are not sufficiently novel.
- Broader testing across varied datasets is expected to test the generalization ability of the method.

In conclusion, while the paper demonstrates some strengths in performance and analytical depth, these do not outweigh the significant issues related to novelty and generalizability. These factors are critical for a contribution to be considered as significant enough in ICLR. After careful discussion and consideration, we regret to inform that this paper is not accepeted in this form.

**Additional Comments On Reviewer Discussion:**

During the rebuttal period, the primary points of discussion centered around the novelty of the paper and the scope of its experimental validation. All of the three reviewers share concern that novelty is not significant enough. Both Reviewer *dknQ* and Reviewer *PoXh* highlighted the importance of extending the empirical results to more datasets and tasks to strengthen the paper's claims. The authors’ responses focused on justifying their method choices and clarifying the implications. However, they did not substantially address the limitations regarding dataset diversity and empirical breadth, which is a key factor in the overall decision to recommend rejection.

---

### Decision · Program_Chairs · 2025-01-22

Reject